

# Proving the 6d Cardy formula
# and matching global gravitational anomalies

**Chi-Ming Chang[1,2]⋆, Martin Fluder[3,4,5]†, Ying-Hsuan Lin[5,6]‡ and Yifan Wang[3,7,8]◦**

**1** Yau Mathematical Sciences Center, Tsinghua University, Beijing, 100084, China
**2** Center for Quantum Mathematics and Physics (QMAP),
University of California, Davis, CA 95616, USA
**3** Joseph Henry Laboratories, Princeton University, Princeton, NJ 08544, USA
**4** Kavli Institute for the Physics and Mathematics of the Universe (WPI),
University of Tokyo, Kashiwa 277-8583, Japan
**5** Walter Burke Institute for Theoretical Physics,
California Institute of Technology, Pasadena, CA 91125, USA
**6** Hsinchu County Environmental Protection Bureau, Hsinchu, Taiwan
**7** Center of Mathematical Sciences and Applications,
Harvard University, Cambridge, MA 02138, USA
**8** Jefferson Physical Laboratory, Harvard University, Cambridge, MA 02138, USA

⋆ wychang@ucdavis.edu, † mfluder@princeton.edu,
‡ yhlin@caltech.edu, ◦ yifanw@princeton.edu

## Abstract

A Cardy formula for 6d superconformal field theories (SCFTs) conjectured by Di Pietro and Komargodski in [1] governs the universal behavior of the supersymmetric partition function on $S^1_\beta \times S^5$ in the limit of small $\beta$ and fixed squashing of the $S^5$. For a general 6d SCFT, we study its 5d effective action, which is dominated by the supersymmetric completions of perturbatively gauge-invariant Chern-Simons terms in the small $\beta$ limit. Explicitly evaluating these supersymmetric completions gives the precise squashing dependence in the Cardy formula. For SCFTs with a pure Higgs branch (also known as very Higgsable SCFTs), we determine the Chern-Simons levels by explicitly going onto the Higgs branch and integrating out the Kaluza-Klein modes of the 6d fields on $S^1_\beta$. We then discuss tensor branch flows, where an apparent mismatch between the formula in [1] and the free field answer requires an additional contribution from BPS strings. This "missing contribution" is further sharpened by the relation between the fractional part of the Chern-Simons levels and the (mixed) global gravitational anomalies of the 6d SCFT. We also comment on the Cardy formula for 4d $\mathcal{N} = 2$ SCFTs in relation to Higgs branch and Coulomb branch flows.

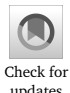



# 1 Introduction

Universal features provide key checks of dualities and reliable handles on otherwise strongly-interacting non-perturbative phenomena. It is long understood that the high temperature/energy limit of quantum field theories exhibits universality. In particular, the Cardy formula for 2d conformal field theories (CFTs) [2] relates the torus partition function, which counts

operators (with refinement), to the Weyl anomaly coefficient (or Virasoro central charge) $c$ in the high temperature limit[1]

$$\log \mathcal{Z}_{S^1_\beta \times S^1} = \frac{\pi^2 c}{3\beta} + \mathcal{O}(\beta^0, \log \beta). \tag{1.1}$$

Here, $\beta$ is the circumference of one of the circles of the torus, while the other is of unit radius. The $\mathcal{O}(\log \beta)$ is due to the potential "noncompactness" of the CFT. For a holographic theory, this formula maps to the universality of the Bekenstein-Hawking entropy of BTZ black holes [3].

In [1], Di Pietro and Komargodski introduced analogs of the Cardy formula for 4d and 6d superconformal field theories (SCFTs), establishing universal relations between perturbative anomalies and the Cardy limit (high temperature limit) of the superconformal index.[2] These higher-dimensional Cardy formulae involve a sum of terms that depend on the background geometry and gauge fields, whose overall coefficients are determined by the perturbative anomaly coefficients. Supersymmetry plays a key role in their higher dimensional Cardy formulae, particularly because supersymmetric partition functions are (expected to be) *geometric invariants* – roughly speaking, quantities that depend only on a subset of the bosonic background data [9–13]. Let us elaborate this point. Unlike the partition function of a 2d CFT on an arbitrary Riemann surface, which only depends on the complex structure (in the absence of chemical potentials), the metric dependence for the partition function of higher dimensional CFTs is much more complicated in general. However, the supersymmetric partition function $\mathcal{Z}_{\mathcal{M}_d}$ for a SCFT on a spacetime manifold $\mathcal{M}_d$ is (believed to be) sensitive to a much smaller subset of the metric data; in the case of even dimensions, this subset comprises the topology and the complex structure moduli of $\mathcal{M}_d$.[3] Taking $\mathcal{M}_d$ to be $S^1 \times \mathcal{M}_{d-1}$, the complex structure dependence of $\mathcal{Z}_{\mathcal{M}_d}$ translates into the dependence on the "transversely holomorphic foliation" structure of $\mathcal{M}_{d-1}$. In fact, the sole dependence of $\mathcal{Z}_{S^1 \times \mathcal{M}_{d-1}}$ on the transversely holomorphic foliation structure of $\mathcal{M}_{d-1}$ has been proven for 3d $\mathcal{N} = 2$ theories in [11, 12] and conjectured for 5d $\mathcal{N} = 1$ theories in [14].[4]

The 4d Cardy formula was proven in [1] for Lagrangian theories continuously connected to free theories via renormalization group (RG) flows triggered by marginal or relevant deformations (moduli flows were not considered, which we remedy in this paper). They evaluated the 3d geometric invariants on squashed sphere backgrounds (the evaluation on lens space was later done in [15]), and determined the coefficients from the Kaluza-Klein (KK) reduction of the free theory. The 6d Cardy formula was first conjectured in [1], and further evidence for the conjecture from the superconformal indices of free theories was found in [1, 16, 17].

In the first part of this paper, we derive the universal background dependence of 6d SCFTs on squashed $S^5$ in the Cardy limit, by studying the 5d effective action from the reduction on $S^1$. Throughout, we make use of various effective actions, whose relations are summarized in Figure 1. A key ingredient in the proof of the 6d Cardy formula is the classification of the 5d supersymmetric Chern-Simons terms in [18] by the present authors, and their explicit forms found earlier in [19–24]. We evaluate these 5d supersymmetric Chern-Simons terms on the most general supersymmetric squashed $S^5$ background, and produce the conjectured expression of [1]. Along the way, we find further evidence for these supersymmetric Chern-Simons terms to be geometric invariants.

---

[1]The convention for the torus moduli is $\tau = \mathrm{i}\frac{\beta}{2\pi}$.

[2]There are other universal limits of the partition function of CFTs in $d > 2$. The governing formulae are also called Cardy formulae in the literature. See *e.g.* [4–8].

[3]This is proven for 4d $\mathcal{N} = 1$ SCFTs in [11, 12], and it would be interesting to pursue a similar argument in 6d by classifying $Q$-exact background couplings.

[4]In [11, 12], it was shown that supersymmetry further demands such a background dependence to be *holomorphic*. Recently, in [13], it was pointed out that this statement is actually (slightly) scheme dependent. Nonetheless, since the different regularization schemes are related by local counter-terms, which are of order $\mathcal{O}(\beta)$ [13], they will not affect the singular terms in the Cardy limit.

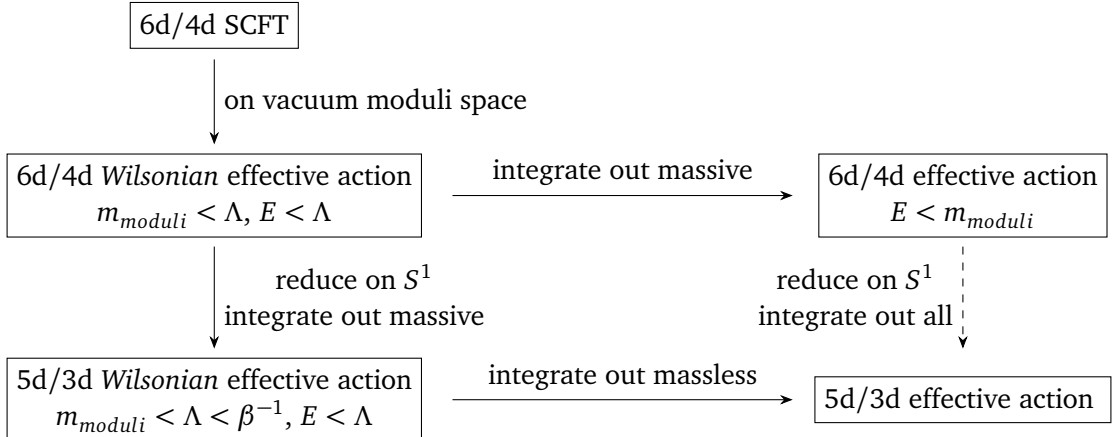

Figure 1: Diagram depicting the relations among different effective actions. We denote by $\Lambda$ the cutoff of the effective action, and $m_{moduli}$ is the mass scale associated with the moduli scalar vacuum expectation value. The field contents are as follows: the *Wilsonian* effective action contains light dynamical fields and background fields; the 6d/4d effective action contains massless dynamical fields and background fields; the 5d/3d effective action contains only background fields. On the one hand, the $\downarrow\rightarrow$ direction is in principle correct but difficult to carry out (unless the effective theory is weakly coupled). On the other hand, except on the pure Higgs branch, where the Green-Schwarz/Wess-Zumino type terms relevant for (mixed) gravitational anomalies are absent in the 6d/4d effective action, it is not understood how to perform the dashed arrow on the right. This is the source of the puzzle of Section 3.3.

In the second part, we determine the Chern-Simons levels that appear in the Cardy formula for a specific SCFT. This is considerably harder in 6d than in 4d, because 6d SCFTs do not have marginal or relevant supersymmetry preserving operator deformations, and there are no known interacting 6d SCFTs with weakly-coupled (UV) Lagrangian descriptions [25, 26]. Fortunately, we can consider moduli space flows. Indeed, for theories with a pure Higgs branch (*i.e.* very Higgsable theories in the language of [27]), where the effective theory far away from the origin is particularly simple and described by free hypermultiplets, we can determine these Chern-Simons levels exactly. This gives a proof of the 6d Cardy formula for very Higgsable SCFTs, which include in particular rank-$N$ E-string theories [28, 29] and $(G, G)$-type minimal conformal matter theories [30].

A similar procedure applied to 4d extends the validity of the 4d proof of [1] to a larger class of theories that includes various non-Lagrangian Argyres-Douglas type theories [31, 32]. We also comment on moduli space flows on the tensor and mixed branches in 6d, and on the Coulomb and mixed branches in 4d.

## 1.1 Review of the 4d Cardy formula

For any $\mathcal{N} \geq 1$ SCFT in 4d, we define the supersymmetric $S_\beta^1 \times S^3$ partition function (or superconformal index) $\mathcal{Z}_{S_\beta^1 \times S^3}$ by [33–36],[5]

$$\mathcal{Z}_{S_\beta^1 \times S^3} = \mathrm{Tr}_{\mathcal{H}}\left[(-1)^F e^{-\hat{\beta}\{Q, Q^\dagger\}} e^{-\beta \sum_{i=1}^2 \omega_i(j_i + R)}\right], \tag{1.2}$$

---

[5]For simplicity, we neglect the fugacities for possible flavor symmetries of the system in this subsection.

where $\mathrm{Tr}_{\mathcal{H}}$ denotes the trace over the Hilbert space $\mathcal{H}$ on $S^3$ in radial quantization, and $j_1$, $j_2$ and $R$ are the Cartan generators of the Lorentz and $U(1)_R$-symmetry groups.[6] The supercharge $Q$ and its conjugate $Q^\dagger$ generate an $\mathfrak{su}(1|1)$ subalgebra of the full $\mathcal{N}=1$ superconformal algebra. The combinations $j_i + R$ commute with the $\mathfrak{su}(1|1)$ and pair with the chemical potentials (squashing parameters) $\omega_i$ that refine and regularize the sum. Due to supersymmetry, only states annihilated by $Q$ and $Q^\dagger$ contribute to the partition function, and consequently the $\hat{\beta}$ dependence drops out. In the Cardy limit $\beta \to 0$, this supersymmetric partition function has the expansion (we fix the radius of $S^3$ to be $r_3 \equiv 1$)

$$\log \mathcal{Z}_{S^1_\beta \times S^3} = \frac{\pi^2}{6\beta} \frac{\omega_1 + \omega_2}{\omega_1 \omega_2} \kappa + \mathcal{O}(\beta^0, \log \beta). \tag{1.3}$$

The coefficient $\kappa$ is related to the pertuburtive mixed gravitational-R-symmetry anomaly, which appears in the anomaly polynomial 6-form as

$$I_6 \ni \frac{k}{48(2\pi)^3} F_R \wedge \mathrm{tr}(R \wedge R), \tag{1.4}$$

with $R$ the Riemann curvature 2-form and $F_R$ the field strength of the background $U(1)_R$ gauge field $V_R$. The relation between $\kappa$ and the anomaly coefficient $k$ is $\kappa = -k$.[7] By supersymmetry, $\kappa$ and $k$ are in turn related to the 4d conformal anomalies as

$$\kappa = -k = 16(c - a). \tag{1.5}$$

This provides a universal relationship between perturbative anomalies and the spectrum of protected BPS operators, and has been explicitly checked in examples by localization computations [1, 15, 43, 44]. The combination of (1.3) and (1.5) is dubbed the 4d Cardy formula, due to its similarity to the 2d Cardy formula (1.1).[8]

In [1], Di Pietro and Komargodski proved the 4d Cardy formula by considering the 4d SCFT compactified on $S^1_\beta$, which leads to a 3d effective action for a set of background fields that include the 3d metric, the background graviphoton gauge field $A$, and the R-symmetry background gauge field $V_R$. To preserve supersymmetry, the fermionic degrees of freedom are chosen to have periodic boundary condition along the $S^1_\beta$. Thus, the 3d effective action contains non-local terms due to integrating out certain massless modes. However, since these modes are uncharged under the Kaluza-Klein $U(1)_{KK}$ symmetry, such non-local terms are sub-

---

[6]The superconformal index differs from the supersymmetric partition function by a Casimir factor [37–40], which vanishes in the Cardy limit $\beta \to 0$. The two are often used interchangeably in this paper, since we are only concerned with the singular terms. Furthermore, one can replace the three-sphere more generally with Seifert manifolds [9, 41, 42].

[7]See footnote 9 for a caveat.

[8]In 4d $\mathcal{N}=2$ SCFTs, an analogous Cardy formula was derived for the Schur index [44] (see also [45])

$$\log \mathcal{Z}^{\mathrm{Schur}}_{S^1_\beta \times S^3} = \frac{\pi^2 \kappa}{2\beta} + \mathcal{O}(\beta^0, \log \beta), \tag{1.6}$$

which also naturally arises in the Cardy limit of the associated 2d chiral algebra [46]. The relative factor of $\frac{3}{2}$ compared to (1.3) in the case $\omega_i = 1$ is due to the ratio between the $U(1)_R$ backgrounds that are turned on in the Schur limit for $\mathcal{N}=2$ SCFTs versus the one for a generic $\mathcal{N}=1$ SCFT, i.e.

$$V^{\mathrm{Schur}}_{U(1)_R} = \frac{3}{2} V_{U(1)_R}. \tag{1.7}$$

Note that the Schur limit also involves turning on the $SU(2)_R$ background gauge field along $S^1_\beta$ in the $\mathcal{N}=2$ SCFT, but this does not affect the Cardy limit.

leading $\mathcal{O}(\beta^0, \log \beta)$ in the $\beta \to 0$ limit.[9] The leading term in the small $\beta$ expansion of the effective action is a Chern-Simons term of order $\mathcal{O}(\beta^{-1})$,

$$i W \;=\; -\log \mathcal{Z} \;=\; \frac{i\kappa}{24\pi} \left( - \int V_R \wedge dA + \text{ SUSY completion} \right) + \mathcal{O}(\beta^0, \log \beta), \qquad (1.8)$$

where the graviphoton $A$ is of order $\mathcal{O}(\beta^{-1})$.[10] Evaluating this supersymmetric effective action on the squashed three-sphere background, one produces the Cardy formula (1.3). What remains is to establish the relation (1.5) between $\kappa$ and the anomaly coefficient $k$.

Let us first provide a qualitative explanation of the relation (1.5) before proving it. In canonical normalization, the Chern-Simons term in (1.8) has level $\frac{\kappa}{12}$, which is in general fractional. Consequently, under a large background $U(1)_{KK}$ gauge transformation, the partition function will pick up a phase. This is nothing but a manifestation of the global (mixed) gravitational anomaly of the 4d CFT [48–50].

Proceeding with the proof, the coefficient $\kappa$ first must be invariant under both marginal and relevant deformations. Otherwise, by promoting the coupling constants of the deformations to background fields, the Chern-Simons term becomes gauge non-invariant under *small* background gauge transformations (which contradicts the absence of such perturbative anomalies in 4d). In particular, this argument forbids not only the continuous dependence of $\kappa$ on the coupling constants, but also potential jumps of $\kappa$, since this would lead to gauge non-invariant domain wall configurations (for the coupling) after promoting the coupling constants to background fields.[11]

Now, let us assume that our 4d SCFT is connected to a weakly coupled theory by either marginal or irrelevant operators that preserve $U(1)_R$.[12] This happens, for example, when the SCFT is located on the conformal manifold of an $\mathcal{N} = 1$ conformal gauge theory or as the IR fixed point of an asymptotically-free gauge theory.[13] Given such a free-field point, one can explicitly KK-reduce the free fields along the $S^1_\beta$. The coefficient $\kappa$ receives contributions from the massive KK-modes via 1-loop diagrams. Summing over such contributions under the appropriate regularization, one finds the relation (1.5) between the coefficient $\kappa$ and the mixed gravitational-R anomaly coefficient $k$, computed at the free point. Since both $\kappa$ and $k$ are invariant under RG flows triggered by deformations in the action, the relation (1.5) holds at the superconformal fixed point as well.

## 1.2 The 4d Cardy formula from moduli space flows

The argument in the previous section can be extended to SCFTs connected to free theories by moduli space flows. Given a 4d SCFT with a vacuum moduli space, generic $S^1_\beta$ reductions would lift these flat directions. However, for *supersymmetric* (Scherk-Schwarz type) $S^1_\beta$ reductions which we consider here, the moduli space is (partly) preserved (and sometimes

---

[9]Strictly speaking, this is only true if the effective action of the 3d massless modes has a minimum at the origin upon turning on general chemical potentials. In particular, there are counter-examples when $c - a < 0$ in which case the $1/\beta$ term in the Cardy limit gets shifted by a non-universal piece due to the existence of nontrivial minima of the potential for the holonomies around $S^1$ [15, 44, 47]. Here, for simplicity we restrict to theories where this phenomena does not occur.

[10]Compared with the convention in [1], our graviphoton field $A_{\text{here}} = \frac{2\pi}{\beta} a_{\text{there}}$.

[11]A priori, such jumps in $\kappa$ could happen if there are extra massless degrees of freedom charged under $U(1)_{KK}$ at special loci of the coupling space. Since we are assuming $\frac{1}{\beta}$ is the largest scale, this is not possible in the present context.

[12]As explained in [25], there are no relevant deformations of an $\mathcal{N} = 1$ SCFT without breaking the $\mathcal{N} = 1$ supersymmetry and $U(1)_R$ symmetry. Moreover, marginal deformations can only become exactly marginal or marginally irrelevant as explained in [51].

[13]In the latter case, one needs to perform $a$-maximization to identify the superconformal $U(1)_R$ symmetry from a combination of the UV $U(1)_R$ symmetry and flavor symmetries [52].

enhanced) in the 3d theory. In particular, for 4d $\mathcal{N} = 2$ SCFTs, the Higgs branch moduli space is preserved and the Coulomb branch moduli space is enhanced (by line operators wrapping the $S^1$) [53]. However, for 4d $\mathcal{N} = 1$ SCFTs described by IR fixed points of asymptotically free gauge theories, it is known that upon $S^1$ reduction parts of the Coulomb branch are lifted due to Affleck-Harvey-Witten type monopole-instantons [54]. In such cases, we use $\phi$ to denote the directions on the 4d moduli space that are not lifted upon $S^1$ reduction. Since the 4d moduli $\phi$ is unambiguously identified as a subspace of the 3d moduli, we will use the same notation. The 3d *Wilsonian* effective action, which does not involve integrating out the (massless) scalars $\phi$ parametrizing the moduli space (and their superpartners), takes the form of (1.8), but with $\kappa$ promoted to a function $\kappa(\phi)$ of the moduli fields.[14] However, any nontrivial dependence of $\kappa(\phi)$ on $\phi$ is again in violation of background small $U(1)_{\text{KK}}$ gauge invariance, so the coefficient $\kappa(\phi)$ must be *constant* in $\phi$.[15]

From the above argument, one would hope to extract $\kappa$ from a weakly coupled description on the 4d moduli space which may involve massive particles and (extended) solitons coupled to the moduli fields.[16] Since we are interested in the supersymmetric partition function, we only expect BPS states to contribute [55–57]. Consequently, $\kappa$ is simply determined by the spectrum of BPS states and their $U(1)_R$ charges (in the full 4d theory on $S^1 \times S^3$ but prior to integrating out massive fields).

For a 4d $\mathcal{N} = 2$ SCFT with a pure Higgs branch of quaternionic dimension $d_H$, which may not have a UV Lagrangian description, the low energy description is simply given by $d_H$ massless hypermultiplets. There are no other BPS particles on the Higgs branch due to the absence of a scalar central charge, but there can be BPS strings (see Appendix B.1 for a discussion) [58–62]. Recall that the $U(1)_R$ symmetry of the $\mathcal{N} = 1$ subalgebra is related to the $SU(2)_R \times U(1)_r$ symmetry of the $\mathcal{N} = 2$ algebra by

$$R_{\mathcal{N}=1} = \frac{4}{3}I_3 + \frac{1}{3}r_{\mathcal{N}=2}, \tag{1.9}$$

where $I_3$ is the Cartan element of the $SU(2)_R$. Upon supersymmetric $S^1$ reduction, the 3d Higgs branch is identical to the 4d Higgs branch [63]. Far on the Higgs branch, the effective theory is described by $d_H$ free hypermultiplets. Thus, by a free field computation as in [1], one finds that[17]

$$\kappa = -\frac{1}{3}\text{Tr}\,(U(1)_r) = \frac{2d_H}{3}. \tag{1.10}$$

On the other hand, from anomaly matching [64],

$$d_H = 24(c - a), \tag{1.11}$$

which confirms (1.5).

One may worry about potential contributions from the BPS vortex strings [59–62]. For Lagrangian theories, by looking at the Higgs branch localization formulae for the index [65,66], we explicitly see that they do not give additional contributions in the Cardy limit. Combined with the above agreement, we expect this to hold in general.

---

[14]The effective action contains the usual kinetic term of $\phi$ plus higher derivative interactions which are of order $\mathcal{O}(\beta^0, \log\beta)$.

[15]We remark that this argument is not affected by the presence of the path-integral measure for $\phi$, since perturbative anomalies, which could potentially absorb such gauge non-invariant pieces are absent in 3d.

[16]Since an explicit mass term (with a $U(1)_R$ invariant mass parameter) pairs chiral and anti-chiral fermions with the same $U(1)_R$ charge, integrating out such massive particles cannot contribute to the Chern-Simons level $\kappa$. Consequently, $\kappa$ can only receive contributions from massive particles when the $U(1)_R$ symmetry is spontaneously broken, so that the mass parameter could carry nonzero $U(1)_R$ charge.

[17]The $\mathcal{N} = 2$ hypermultiplet contains two left-handed Weyl fermions with $(I_3, r_{\mathcal{N}=2}) = (0, -1)$.

The above argument extends the validity of the 4d Cardy formula to $\mathcal{N} = 2$ SCFTs with a pure Higgs branch, on which there is no additional contribution to $\kappa$ beyond the massless hypermultiplets. In particular, one can explicitly check that it is obeyed by a large class of Argyres-Douglas type theories (for instance) using the relation to 2d chiral algebras [31, 45, 46].[18]

On the contrary, the Coulomb branch is known to host a zoo of BPS particles which undergo complicated decays and recombinations as one explores the moduli space (from one chamber to another), characterized by the wall-crossing phenomena. However, as we have argued, $\kappa(\phi)$ must remain constant. In other words, one can pick any chamber and compute $\kappa$ from the stable BPS spectrum within the given chamber. If the theory has a weakly coupled chamber, such as in Lagrangian theories where the stable BPS particles are W-bosons and massive hypermultiplets, it is easy to see that (1.5) holds.[19] However, a complication arises when there is no weakly coupled chamber. This happens for instance on the Coulomb branch of 4d $\mathcal{N} = 2$ Argyres-Douglas theories, which has mutually non-local BPS monopoles and dyons, and consequently the effective theory is never weakly coupled [31, 32]. In such cases, the interactions between the BPS particles *cannot* be neglected in the Cardy limit and will contribute to $\kappa$.

One may attempt to determine $\kappa$ using a different procedure by first integrating out the massive BPS states in 4d which induces various higher derivative terms for the moduli fields, and then studying the $S^1$ reduction of the effective action that only involves the massless moduli (and the supersymmetric partners). For the supersymmetric partition function, only F-type higher derivative terms contribute (D-terms are necessarily $Q$-exact). This includes the supersymmetric Wess-Zumino term [67–69] (*i.e.* it contains the Wess-Zumino term for the Weyl $a$-anomaly used in [70, 71] to prove the 4d $a$-theorem), which is essential for matching the perturbative mixed anomaly of the CFT on the Coulomb branch involving the spontaneously broken $U(1)_r$ symmetry [72]. When the Coulomb branch is one dimensional, it was argued in [73] that this is the only F-term that can contribute on $S^1 \times S^3$.[20] However, it is not clear to us how the Wess-Zumino terms (and its supersymmetric completion) or other F-terms can contribute to the level $\kappa$. We leave this for future investigation.[21]

## 1.3 6d Cardy formula

Now, let us consider a 6d $\mathcal{N} = (1,0)$ SCFT in radial quantization. The states in the Hilbert space are labeled by the energy $E$, the spins $j_1, j_2, j_3$ of the $SO(2)^3 \subset SO(6)$ rotation of the $S^5$, and the R-charge $R$ of the $SU(2)_R$ R-symmetry. The theory has eight Poincaré $Q$-supercharges $Q_{j_1,j_2,j_3,R}$, and eight superconformal cousins. In principle, the full spectrum of this theory is encoded in its $S^1 \times S^5$ partition function with anti-periodic boundary condition for the fermions along the $S^1$. However, such a quantity is generally difficult to compute because the fermionic boundary condition breaks supersymmetry.

A more manageable quantity is the superconformal index, which is related by a Casimir factor to the $S^1 \times S^5$ partition function with periodic boundary conditions for the fermions, and with suitable chemical potentials turned on [40, 74]. It is also a counting function, but only for states saturating a BPS bound. To define it, we first pick a particular supercharge $Q \equiv Q_{---+}$,

---

[18]See footnote 8 for an explanation of a relative factor of $\frac{3}{2}$ where comparing the Cardy formulae that appeared in [44, 46] with that in [1].

[19]Note that the massive $\mathcal{N} = 2$ vector multiplet contains two left-handed Weyl fermions with $(I_3, r_{\mathcal{N}=2}) = (\pm 1/2, 1)$, respectively, so each W-boson contributes $-\frac{2}{3}$ to $\kappa$.

[20]In [73], it was proven that for one dimensional Coulomb branches, the only F-terms are proportional to the Weyl tensor and its derivatives which all vanish on $S^1 \times S^3$.

[21]A logical possibility is that (a) integrating out the massive matter in 4d and (b) compactifying on $S^1$ do not commute, and thus the order of first (a) then (b) misses subtle contributions to $\kappa$.

whose anticommutator reads

$$\{Q,Q^\dagger\} = E - j_1 - j_2 - j_3 - 4R \geq 0. \tag{1.12}$$

Given this choice, the states annihilated by the supercharge $Q$ and therefore saturating the BPS bound (1.12) can be counted (with signs) by the superconformal index [74–76] (see also [77] for a review),

$$\mathcal{Z}_{S^1_\beta \times S^5} = \mathrm{Tr}_{\mathcal{H}} \left[ (-1)^F e^{-\hat\beta \{Q,Q^\dagger\} - \beta \sum_I \mu_{\mathrm{f}}^I H_{\mathrm{f}}^I - \beta \sum_{i=1}^3 \omega_i (j_i + R)} \right]. \tag{1.13}$$

Here, the trace is taken over the Hilbert space $\mathcal{H}$ of states on $S^5$, and $\beta$ is the circumference of the $S^1$. Furthermore, we have introduced (complex) chemical potentials $\{\omega_i\}_{i=1}^3$ (with Re $\omega_i > 0$) for the isometries, and $\mu_{\mathrm{f}}^I$ for the flavor symmetries. Finally, $H_{\mathrm{f}}^I$ are the generators of the Cartan subalgebra of the Lie algebra of the flavor symmetry $G_{\mathrm{f}}$.[22]

Di Pietro and Komargodski [1] conjectured a Cardy formula for the Cardy limit $\beta \to 0$ of the superconformal index, stating that the singular terms in this limit take the form

$$
\begin{aligned}
\log \mathcal{Z}_{S^1_\beta \times S^5} = &-\frac{\pi}{\omega_1 \omega_2 \omega_3} \Bigg[ \frac{\kappa_1}{360} \left( \frac{2\pi}{\beta} \right)^3 + \frac{(\omega_1^2 + \omega_2^2 + \omega_3^2)(\kappa_2 - 3\kappa_3/2)}{72} \left( \frac{2\pi}{\beta} \right) \\
&+ \frac{(\omega_1 + \omega_2 + \omega_3)^2 \kappa_3}{48} \left( \frac{2\pi}{\beta} \right) + \frac{\mu_{\mathrm{f}}^2 \kappa_{\mathrm{f}}^{G_{\mathrm{f}}}}{24} \left( \frac{2\pi}{\beta} \right) \Bigg] + \mathcal{O}(\beta^0, \log\beta),
\end{aligned}
\tag{1.14}
$$

with the constants $\kappa_i$ completely fixed by the perturbative anomalies as follows.[23] We write the 8-form anomaly polynomial as

$$I_8 = \frac{1}{4!} \left[ \alpha c_2(SU(2)_R)^2 + \beta c_2 p_1 + \gamma p_1^2 + \delta p_2 \right] + \mu^{G_{\mathrm{f}}} p_1 c_2(G_{\mathrm{f}}), \tag{1.15}$$

where the explicit formulae for the Chern classes $c_i$ and Pontryagin classes $p_i$ in terms of the field strengths and curvature are summarized in Appendix A. Here, $G_{\mathrm{f}}$ is the flavor symmetry of the 6d SCFT. Then, the relations are

$$\kappa_1 = -40\gamma - 10\delta, \quad \kappa_2 - \frac{3}{2}\kappa_3 = 16\gamma - 2\delta, \quad \kappa_3 = -2\beta, \quad \kappa_{\mathrm{f}}^{G_{\mathrm{f}}} = -48\mu^{G_{\mathrm{f}}}. \tag{1.16}$$

### $\mathcal{N} = (2,0)$ theory check

Before pursuing a proof, let us explicitly check the 6d Cardy formula in $\mathcal{N} = (2,0)$ theories where we have a closed form expression for the superconformal index in the limit $\omega_1 = \omega_2 = \omega_3 = 1$ (unsquashed) and $\mu_{\mathrm{f}} = 1$, computed by localization in [74,76]. The result for the theory of type-$\mathfrak{g}$, where $\mathfrak{g}$ is a simply laced simple Lie algebra, is

$$\mathcal{Z}_{S^1_\beta \times S^5}^{\mathfrak{g}} = e^{\frac{\beta}{6} h_{\mathfrak{g}}^\vee |\mathfrak{g}|} \left( \frac{\beta}{2\pi} \right)^{\frac{r_{\mathfrak{g}}}{2}} \prod_{\alpha \in \Delta_+^{\mathfrak{g}}} \left( 1 - e^{-\beta(\alpha \cdot \rho_{\mathfrak{g}})} \right) \eta\left( e^{-\frac{4\pi^2}{\beta}} \right)^{r_{\mathfrak{g}}}, \tag{1.17}$$

where $h_{\mathfrak{g}}^\vee$ is the dual Coxeter number, $r_{\mathfrak{g}}$ is the rank, $|\mathfrak{g}|$ is the dimension, $\Delta_+^{\mathfrak{g}}$ is the set of positive roots, and $\rho_{\mathfrak{g}}$ is the Weyl vector. Notice that the result for $\mathfrak{g} = E$ is only conjectural, as

---

[22]The normalization of $H_{\mathrm{f}}^I$ is chosen to be $\mathrm{Tr}(H_{\mathrm{f}}^I H_{\mathrm{f}}^J) = \delta^{IJ}$.

[23]As in the 4d/3d case, we assume that the 5d effective action of the massless modes has a minimum at the origin as $\beta \to 0$ (see footnote 9). We checked this explicitly when the 6d index has a known matrix model description, *i.e.* for $\mathcal{N} = (2,0)$ theories and E-string theories.

the instanton contributions are unknown. In the Cardy limit, the last factor of (1.17) becomes the dominant contribution, and we find

$$\log \mathcal{Z}^{\mathfrak{g}}_{S^1_\beta \times S^5} = \frac{r_\mathfrak{g} \pi^2}{6\beta} + \mathcal{O}(\beta^0, \log \beta). \tag{1.18}$$

To compare, the Cardy formula (1.14) with the Chern-Simons levels $\kappa_i$ given in Table 2 dictates that the $S^1_\beta \times S^5$ partition function in the Cardy limit is

$$\log \mathcal{Z}_{S^1_\beta \times S^5} = \frac{r_\mathfrak{g} \pi^2}{24\beta \omega_1 \omega_2 \omega_3}(2\omega_1\omega_2 + 2\omega_2\omega_3 + 2\omega_3\omega_1 + \mu_\mathrm{f}^2 - \omega_1^2 - \omega_2^2 - \omega_3^2) + \mathcal{O}(\beta^0, \log \beta), \tag{1.19}$$

which with $\omega_1 = \omega_2 = \omega_3 = 1$ (unsquashed) and $\mu_\mathrm{f} = 1$ *matches* with the localization result (1.18).

The $\beta \to 0$ limit of the type-$\mathfrak{g}$, $\mathcal{N} = (2,0)$ superconformal index commutes with the "chiral algebra limit" of the 6d theory in which its superconformal index reduces to the vacuum character of a 2d $\mathcal{W}_\mathfrak{g}$ algebra [78]. This implies that the Cardy formula of the corresponding 2d VOA coincides with the 6d (supersymmetric unsquashed) Cardy formula (1.18), and the modular properties of the characters lead to a high/low-temperature relation in the 6d parent between the Casimir energy and the Cardy limit. See [44, 46] for analogous statements in the 4d case.

## 1.4 Sketch of the proof

We presently outline our proof of the 6d Cardy formula, given by (1.14) and (1.16), for theories with (at least) a pure Higgs branch. The first step is to analyze the most general 5d effective action of 6d SCFTs compactified on $S^1_\beta$. As argued in [1] and in Section 3, the 5d effective action has an expansion in the small radius $\beta \to 0$ limit as

$$\mathrm{i}W = -\log \mathcal{Z} = \frac{\mathrm{i}}{8\pi^2}\left(\frac{\kappa_1}{360}I_1 + \frac{\kappa_2 - \frac{3}{2}\kappa_3}{144}I_2 - \frac{\kappa_3}{24}I_3 - \frac{\kappa_\mathrm{f}^{G_\mathrm{f}}}{24}I_4^{G_\mathrm{f}}\right) + \mathcal{O}(\beta^0, \log \beta), \tag{1.20}$$

which contains four types of supersymmetric Chern-Simons terms

$$I_1 \equiv \int A \wedge \mathrm{d}A \wedge \mathrm{d}A + \text{ SUSY completion},$$

$$I_2 \equiv \int A \wedge \mathrm{tr}(R \wedge R) + \text{ SUSY completion},$$

$$I_3 \equiv \int A \wedge \mathrm{Tr}(F_R \wedge F_R) + \text{ SUSY completion},$$

$$I_4^{G_\mathrm{f}} \equiv \int A \wedge \mathrm{Tr}(F_{G_\mathrm{f}} \wedge F_{G_\mathrm{f}}) + \text{ SUSY completion}. \tag{1.21}$$

Here, $A$ is the $U(1)_\mathrm{KK}$ graviphoton (which in the $\beta \to 0$ limit scales as $\beta^{-1}$), $R$ denotes the Riemann curvature 2-form of the 5d background metric $h_{ij}$,

$$\mathrm{ds}_6^2 = \left(\mathrm{d}\tau + \frac{\beta}{2\pi}A_i\mathrm{d}x^i\right)^2 + h_{ij}\mathrm{d}x^i\mathrm{d}x^j, \tag{1.22}$$

$F_R$ is the field strength of the $SU(2)_R$ background gauge field, and lastly $F_{G_\mathrm{f}}$ is the field strength of background flavor symmetry gauge fields.[24] The key realization is that only these supersymmetric Chern-Simons terms contribute to singular terms in the $\beta \to 0$ limit.

---

[24]Recall that on the supersymmetric background, the $SU(2)_R$ gauge field takes value in the Cartan of $SU(2)_R$ (see Appendix C).

The squashing dependence of the 6d Cardy formula (1.14) is recovered by evaluating the effective action (1.20) on the rigid supersymmetric background of a squashed $S^5$, with the squashing parameters $\omega_i = 1 + a_i$ of the 5d metric (see Section 2.1). We stress that the contributions from the additional supersymmetric pieces in (1.21) are absolutely crucial to our result; without them, the result is *not* a geometric invariant depending only on the squashing parameters.

The second part of the 6d Cardy formula is the relation (1.16) between the Chern-Simons levels $\kappa_i$ in (1.14) and the perturbative anomaly coefficients. To derive this relation, we first argue that the Chern-Simons levels are *constant* on the vacuum moduli space, similar to our argument in Section 1.1 for the 4d case. Consider the 5d *Wilsonian* effective action that does not involve integrating out the (massless) moduli scalars $\phi$ (and the superpartners). It takes the form (1.20) but with the Chern-Simons levels $\kappa_i$ promoted to functions $\kappa_i(\phi)$ of the moduli fields. Note that the scalars descend to moduli fields in the 5d theory, since these flat directions are not lifted under supersymmetric $S^1_\beta$ reduction. Any nontrivial dependence of $\kappa(\phi)$ on $\phi$ is in violation of background small gauge invariance, so the coefficients $\kappa_i(\phi)$ must be *constant* on the entire vacuum moduli space.

For 6d SCFTs with a pure Higgs branch, on which we just have several massless hypermultiplets and no other BPS particles (see Appendix B.2 for a discussion), the above argument allows us to determine the Chern-Simons levels by simply reducing free fields. This procedure proves the relation (1.16) for theories possessing a pure Higgs branch, with the understanding that $\gamma = -\frac{7}{4}\delta$, which is inherently true for such theories.

To derive the relation (1.16) on the tensor branch, we compute the one-loop contributions from the free fields to the Chern-Simons levels $\kappa$, and find that there *must* be additional contributions. It is known that the 6d tensor branch supports BPS strings which may contribute to $\kappa$ upon reduction on $S^1$. Indeed, this is evident from the conjectured localization formulae for the $S^1 \times S^5$ partition function of 6d SCFTs [74, 76, 77, 79] (see also Section 3.3). However, it is not clear how to systematically include contributions from such states to $\kappa$. Instead, if we first integrate out the massive states on the 6d tensor branch, we obtain a tower of higher derivative interactions in the effective action. In particular, one of the leading F-terms is given by (the supersymmetric completion of) the Green-Schwarz term. Then, one needs to study the contribution from such terms to $\kappa$ upon reduction on $S^1$; we will not pursue this in the present paper.

To provide another perspective on the would-be contributions from BPS strings to $\kappa$, we consider the relation between the Chern-Simons levels in the 5d effective action and the global anomalies of the 6d SCFT, and explain that the extra contributions are essential for the global anomaly matching. The global gravitational anomalies of the 6d SCFT can receive contributions from the Green-Schwarz term in the tensor branch effective action [80–86] (just as for perturbative anomalies), therefore we expect that such terms are responsible for additional contributions to the 5d Chern-Simons levels upon compactification on $S^1$.

The remainder of this paper is organized as follows. In Section 2, we prove the squashing dependence of the 6d Cardy formula, by solving for appropriate 5d backgrounds with rigid supersymmetry, and evaluating the supersymmetric Chern-Simons terms on such backgrounds. In Section 3, we fix the Chern-Simons levels in terms of the perturbative anomaly coefficients, by combining non-renormalization arguments and free field computations on the pure Higgs branch. An analogous computation on the pure tensor branch is then performed which yields a naive mismatch with (1.14) but the "missing" contributions must come from the BPS strings. In Section 4, we remark on the relation between the Chern-Simons levels and global gravitational anomalies, which further highlights the "missing" contributions on the tensor branch. Section 5 closes with a summary and comments on future directions. Finally, in the four appendices we provide more details on various aspects discussed in the main text. In particular,

in Appendix C we detail the 6d supersymmetric background on $S^1 \times S^5$, including its "modified/complexified version" and how it relates to our 5d backgrounds.

# 2 Supersymmetric Chern-Simons terms on the squashed $S^5$ background

In this section, we study the 5d *supersymmetric* Chern-Simons terms (1.21) and their supersymmetric completions arising from the dimensional reduction of the 6d theory, and evaluate them on the squashed $S^5$ background. These Chern-Simons terms correspond to the higher-derivative terms in 5d Poincaré supergravity classified in [18].

We begin with the $S^1 \times S^5$ geometric background and reduce to the 5d background that also includes a graviphoton field, and further embed this bosonic background into supergravity to obtain the full 5d supersymmetric background. We review various ingredients, *i.e.* various (matter) multiplets and a version of Poincaré supergravity that arises from a particular choice of gauge-fixing of the (conformal) standard Weyl multiplet together with a vector and a linear multiplet.[25] Then, following the general formalism of [36], we provide the rigid supersymmetric background for a generically squashed $S^5$. Finally, we detail the corresponding higher-derivative terms (supersymmetric Chern-Simons terms) and their evaluation on the supersymmetric squashed $S^5$ background.

Readers who are not interested in the technicalities of 5d supersymmetric solutions for various multiplets may skip this section and consult the results summarized in Section 2.5.

## 2.1 Squashed $S^5$ background from reduction of squashed $S^1 \times S^5$ background

We begin by specifying the appropriate $S^1 \times S^5$ background of the 6d theory. This can be done most straightforwardly by employing a conformal transformation from flat $\mathbb{R}^6$ to $\mathbb{R} \times S^5$, and then compactifying the non-compact direction with twisted boundary conditions for various fields induced by the chemical potentials in (1.13). The introduction of such chemical potentials reduces the amount of preserved supercharges. We absorb the twists for the isometries of the $S^5$ into the geometry (for the sake of setting some background fields to zero), and fix the (deformed) metric of $S^1 \times S^5$ to be

$$\mathrm{d}s^2_{S^1 \times S^5} = r_5^2 \sum_{i=1}^{3} \left[ \mathrm{d}y_i^2 + y_i^2 \left( \mathrm{d}\phi_i + \frac{\mathrm{i}a_i}{r_5} \mathrm{d}\tau \right)^2 \right] + \mathrm{d}\tau^2 , \tag{2.1}$$

where $\tau \sim \tau + \beta$ is the $\beta$-periodic $S^1$ coordinate, and $\{y_i, \phi_i\}_i$ are polar coordinates, satisfying $y_1^2 + y_2^2 + y_3^2 = 1$ as well as $\phi_i \sim \phi_i + 2\pi$. Finally, $r_5$ is the radius of the five-sphere, and the chemical potentials $\omega_i$ are related to the metric deformations via

$$\omega_i = 1 + a_i , \tag{2.2}$$

the round case being $\omega_i = 1$. The twisting parameters $a_i$ can in general be complex numbers. The full 6d supersymmetric background and the background for the "modified" index are detailed in Appendix C.

As we are interested in the compactification to five dimensions, we can conveniently rewrite the 6d metric as follows

$$\mathrm{d}s^2_{S^1 \times S^5} = \tilde{\kappa}^{-2} (\mathrm{d}\tau + \mathrm{i}r_5 \mathcal{Y})^2 + \mathrm{d}s_5^2 , \tag{2.3}$$

---

[25]Note that this is a different choice of Poincaré supergravity than the one employed in [87].

with the (squashed) $S^5$ metric $\mathrm{d}s_5^2$,

$$\mathrm{d}s_5^2 = \sum_{i=1}^{3}\left(\mathrm{d}y_i^2 + y_i^2 \mathrm{d}\phi_i^2\right) + \tilde{\kappa}^{-2}\mathcal{Y}^2,$$

$$\mathcal{Y} = \tilde{\kappa}^2 \sum_{i=1}^{3} a_i y_i^2 \mathrm{d}\phi_i, \tag{2.4}$$

$$\tilde{\kappa}^{-2} = 1 - \sum_{j=1}^{3} y_j^2 a_j^2.$$

When the twisting parameters $a_i \in \mathbb{R}$, they correspond to the squashing parameters of the squashed $S^5$ ($a_i = 0$ corresponds to the round $S^5$ limit). A comparison of the metrics (1.22) and (2.3) (up to a conformal transformation) shows that the graviphoton field $A$ in the squashed $S^5$ background is

$$A = m_{\mathrm{KK}} r_5 \mathcal{Y}, \qquad m_{\mathrm{KK}} = \frac{2\pi \mathrm{i}}{\beta}. \tag{2.5}$$

In the following, we set $r_5 = 1$.

## 2.2 Off-shell 5d supergravity multiplets

The 5d higher-derivative supersymmetric Chern-Simons terms can be written in terms of various 5d (off-shell) supergravity multiplets. On the one hand, we work with the standard Weyl multiplet coupled to vector multiplets [19, 20]; on the other hand, we work with the 5d off-shell Poincaré supergravity. The former is enough if the Chern-Simons term preserves conformal invariance, which in Poincaré supergravity is explicitly broken. The Poincaré supergravity is obtained by gauge-fixing the (conformal) standard Weyl multiplet [19, 20, 88, 89]. In the following, we pick a particular gauge-fixing condition which naturally reduces the $\mathfrak{su}(2)_R$ symmetry of the standard Weyl multiplet to its $\mathfrak{u}(1)_R$ truncation, which is convenient because there is a general description of 5d rigid supersymmetric backgrounds of (the $\mathfrak{u}(1)_R$ truncation of) the standard Weyl multiplet [14, 90].

We review the various multiplets of interest, before presenting the solution on the squashed $S^5$ in the next section.

**Standard Weyl multiplet**

The (full) 5d standard Weyl multiplet consists of the following matter content

$$\mathcal{SW} = \left(g_{\mu\nu}, D, V_\mu^{ij}, v_{\mu\nu}, b_\mu, \psi_\mu^i, \chi^i\right), \tag{2.6}$$

given by the metric $g_{\mu\nu}$, a dilaton $D$, an $\mathfrak{su}(2)_R$ gauge field $V_\mu^{ij}$, a two-form field $v_{\mu\nu}$, a gauge field $b_\mu$ of the Weyl symmetry, and two $\mathfrak{su}(2)$ Majorana fermions $\psi_\mu^i$ and $\chi^i$. To obtain a rigid supersymmetric background [36], we have to set the fermions to zero and thus find (at least) one non-vanishing spinors such that the supersymmetry variation of the fermions vanish. In particular, for a generic transformation

$$\delta = \bar{\varepsilon}^i Q_i + \bar{\eta}^i S_i, \tag{2.7}$$

where $Q_i$ and $S_i$ are the supercharges and their conformal cousins, and $\varepsilon^i$, $\eta^i$ are the respective parameters specifying the transformation, the supersymmetry variation of the fermions of the standard Weyl multiplet reads

$$\delta \psi_\mu^i = D_\mu \varepsilon^i + \frac{1}{2} v^{\nu\rho} \gamma_{\mu\nu\rho} \varepsilon^i - \gamma_\mu \eta^i,$$

$$\delta \chi^i = \varepsilon^i D - 2\gamma^\rho \gamma^{\mu\nu} \varepsilon^i \nabla_\mu v_{\nu\rho} + \gamma^{\mu\nu} F_{\mu\nu}{}^i{}_j(V)\varepsilon - 2\gamma^\mu \varepsilon^i \epsilon_{\mu\nu\rho\sigma\lambda} v^{\nu\rho} v^{\sigma\lambda} + 4\gamma^{\mu\nu} v_{\mu\nu} \eta^i. \tag{2.8}$$

Here, $F(V)$ is the field strength of the $\mathfrak{su}(2)_R$ symmetry gauge field $V$. The covariant derivative is given by

$$D_\mu \varepsilon^i = \partial_\mu \varepsilon^i + \frac{1}{2} b_\mu \varepsilon^i + \frac{1}{4} \omega_\mu{}^{ab} \gamma_{ab} \varepsilon^i - V_\mu{}^{ij} \varepsilon_j. \tag{2.9}$$

We recall that the doublet indices $i, j \in \{1, 2\}$ are raised and lowered with the totally antisymmetric tensor $\epsilon^{ij}$ using NW-SE and SW-NE conventions, i.e.,

$$\zeta^i = \epsilon^{ij} \zeta_j, \qquad \zeta_i = \epsilon_{ij} \zeta^j, \tag{2.10}$$

with $\epsilon^{12} = -\epsilon_{12} = 1$. Notice that $b_\mu$ corresponds to the gauge field for Weyl transformations (and to obtain non-conformal supergravity we have to gauge-fix it). Furthermore, in the present context we shall work with a $\mathfrak{u}(1)$-truncated version of the standard Weyl multiplet – i.e. the gauge field $(V_\mu)^i{}_j$ only has components along the Cartan generator of $\mathfrak{su}(2)_R$ – in which case the supersymmetry conditions (2.8) can be recast in terms of certain simple geometric constraints on the 5d background [14, 90].

**Vector multiplet**

The 5d supergravity vector multiplet contains the fields

$$\mathcal{V} = \left( W_\mu, M, \Omega_\alpha^i, Y^{ij} \right), \tag{2.11}$$

where $W_\mu$ is the gauge field (the background gauge field for the 5d flavor symmetry), $M$ a scalar (the "scalar mass parameter"), $\Omega_\alpha^i$ the gaugino, and $Y^{ij}$ a triplet of auxiliary scalars. In order to obtain a rigid supersymmetric background, we set the fermions to zero, and thus their variation [20, 91]

$$\delta \Omega^i = -\frac{1}{4} \gamma^{\mu\nu} F_{\mu\nu}(W) \varepsilon^i - \frac{1}{2} \slashed{D} M \varepsilon^i + Y^i{}_j \varepsilon^j - M \eta^j \tag{2.12}$$

has to vanish. Here, $F_{\mu\nu}(W)$ denotes the field strength of the vector multiplet gauge field $W$. This version of the vector multiplet is naturally coupled to the standard Weyl multiplet $\mathcal{SW}$, with the same supersymmetry parameters, $\varepsilon^i$ and $\eta^i$.

**Linear multiplet**

As we shall see, in order to describe Poincaré supergravity, we have to add a vector compensator multiplet as well as another either linear or hypermultiplet compensator. We opt to go with the former option, and thus introduce the 5d linear multiplet $\mathcal{L}$ here. It contains the fields

$$\mathcal{L} = \left( L_{ij}, \varphi_\alpha^i, E^\mu, N \right), \tag{2.13}$$

where $N$ and $L_{ij}$ are scalars, $E^\mu$ is a divergence-less vector, i.e.

$$\nabla^\mu E_\mu = 0, \tag{2.14}$$

and $\varphi_\alpha^i$ are their fermionic partners. Again, to preserve rigid supersymmetry, we set the fermions to zero and require their variation [20, 91]

$$\delta \varphi^i = -\slashed{D} L^i{}_j \varepsilon^j + \frac{1}{2} \gamma^\mu \varepsilon^i E_\mu + \frac{1}{2} \varepsilon^i N + 2 \gamma^{\mu\nu} \nu_{\mu\nu} \varepsilon^j L^i{}_j - 6 L^{ij} \eta_j \tag{2.15}$$

to vanish.

**Poincaré supergravity**

Now, to obtain 5d Poincaré supergravity, we add compensators (denoted by hatted symbols) to the standard Weyl multiplet $\mathcal{SW}$ and gauge-fix the conformal symmetry [20, 23, 92]. In the following, we shall pick the compensators to be given by a vector multiplet $\hat{\mathcal{V}}$ and a linear multiplet $\hat{\mathcal{L}}$.[26] To gauge-fix the superconformal transformations and the dilatation one may impose the following "standard gauge" conditions[27]

$$\text{"Standard gauge":} \qquad \hat{M} = m, \quad b_\nu = 0, \quad \hat{L}^i{}_j = \frac{\mathrm{i}}{2}\hat{L}(\sigma_3)^i{}_j, \quad \hat{\Omega}^i = 0. \qquad (2.16)$$

Here, $m$ is (generally chosen to be) an arbitrary constant of mass dimension one, and $b_\nu$ is the gauge field for dilatations. The first constraint fixes dilatations (up to a constant), the second constraint fixes special conformal transformations, the third reduces the $\mathfrak{su}(2)_R$ symmetry down to $\mathfrak{u}(1)_R$, and the last one fixes the $S$-supersymmetry.

There is another way to gauge-fix the Weyl and the superconformal symmetries given by the "KK gauge" condition,[27]

$$\text{"KK gauge":} \qquad \hat{L}^i{}_j = \frac{\mathrm{i}}{2}\hat{L}(\sigma_3)^i{}_j, \quad \hat{L} = 1, \quad b_\nu = 0, \quad \hat{\varphi}^i = 0. \qquad (2.17)$$

As before, the first constraint in (2.17) breaks $\mathfrak{su}(2)_R$ to $\mathfrak{u}(1)_R$, the second and third fixes dilatations and special conformal transformations, respectively, and the last one fixes the $S$-supersymmetry transformation. Note that while $\hat{L}^i{}_j$ in the compensator linear multiplet is completely fixed by the gauge condition, the scalar $\hat{M}$ in the compensator vector multiplet is unfixed; We are free to choose its value to be non-constant and proportional to the warping factor $\tilde{\kappa}$ defined in (2.4),

$$\hat{M} = M_{\text{KK}}, \qquad \text{where} \qquad M_{\text{KK}} = -m_{KK}\tilde{\kappa} \qquad (2.18)$$

is the (fixed) "KK-mass".[28]

The compensator vector multiplets $\hat{\mathcal{V}}$ in these two gauges have different physical meanings. In the standard gauge, $\hat{\mathcal{V}}$ should be interpreted as a background vector multiplet for a Cartan component of the flavor symmetry, whose mass parameter is a constant. In the KK gauge, $\hat{\mathcal{V}}$ should be interpreted as the background vector multiplet for the $U(1)_{\text{KK}}$ symmetry, whose mass parameter is *position-dependent* and determined by the warping parameter $\tilde{\kappa}$.

The two gauge-fixing conditions, (2.16) and (2.17), are related to one-another by a Weyl transformation (prior to gauge-fixing it); we refer to Appendix D for more details. In fact, we explicitly find that evaluating the FRR terms for either gauge-fixing (2.16) or (2.17) leads to the same answer in the case of (generic) squashed $S^5$. This gives more credence to the fact that the answers should be "geometric invariants", *i.e.* only dependent on some general geometric structure (in our case believed to be the transversely holomorphic foliation) and not on the particular choice of background fields.

The fields of the standard Weyl multiplet $\mathcal{SW}$ together with the compensator vector $\hat{\mathcal{V}}$ and linear multiplet $\hat{\mathcal{L}}$, gauge-fixed using the conditions (2.16) make up the full (off-shell) Poincaré supergravity multiplet

$$\mathcal{P}_{\text{std}} = \left( g_{\mu\nu}, D, V_\mu^{ij}, v_{\mu\nu}, \hat{W}_\mu, \hat{Y}^{ij}, \hat{E}^\mu, \hat{L}, \hat{N}, \psi_{\mu\alpha}^i, \chi_\alpha^i, \hat{\varphi}_\alpha^i \right). \qquad (2.19)$$

---

[26]Alternatively, one can add vector multiplets and a hypermultiplet as compensators; this was considered in [21, 88].

[27]The naming of the two gauge-fixing conditions is for the convenience of referencing in this paper, and is not standard in the literature. There are various other ways to fix the conformal symmetry and get different versions of Poincaré supergravity, none of which is distinguished (see *e.g.* [92]).

[28]Since the 6d geometry (2.4) is warped, the KK-mass is not constant but rather proportional to the warping factor $\tilde{\kappa}$ defined in (2.3).

In the case of the "KK gauge" (2.17), we get the following (independent) component fields in the Poincaré multiplet

$$
\mathcal{P}_{\text{KK}} = \left( g_{\mu\nu} \,,\, D \,,\, V_{\mu}^{ij} \,,\, \nu_{\mu\nu} \,,\, \hat{M} \,,\, \hat{W}_{\mu} \,,\, \hat{Y}^{ij} \,,\, \hat{E}^{\mu} \,,\, \hat{N} \,,\, \psi_{\mu\alpha}^{i} \,,\, \chi_{\alpha}^{i} \,,\, \hat{\Omega}_{\alpha}^{i} \right) . \tag{2.20}
$$

In the rest of this paper, we will work in the KK gauge if the compensator gauge field $\hat{W}$ is identified with the graviphoton gauge field $A$. This involves the supersymmetric completion of the $A \wedge \text{tr}\,(R \wedge R)$ term, which contains the graviphoton and is written in terms of Poincaré supergravity. On the other hand, when we evaluate the supersymmetric completion of the $A \wedge \text{Tr}\,(F_{G_f} \wedge F_{G_f})$ term, we have to include some other (non-compensator) background vector multiplets, in which the gauge fields $A_{G_f}$ of the flavor symmetry reside.

To get a supersymmetric background of Poincaré supergravity, we take the rigid limit, and thus set the fermionic fields to zero. Hence, we are required to find non-trivial solutions for the Killing spinors $\varepsilon^i$ to the vanishing fermionic supersymmetry transformations (2.8), (2.12) and (2.15), whilst imposing the gauge-fixing conditions of (2.16) or (2.17).

## 2.3 Squashed $S^5$ backgrounds in off-shell supergravity

To evaluate the 5d supersymmetric Chern-Simons terms on the squashed $S^5$ background (2.4), we consider the rigid limit of the various supergravity multiplets discussed in the previous section. We first present the solution for the standard Weyl multiplet, $\mathcal{SW}$, and then proceed with solving for the flavor vector multiplets, the KK vector multiplets and finally the full (KK-gauge fixed) Poincaré supergravity.

**Solution for the standard Weyl multiplet**

The solution for the bosonic background fields $(V^i{}_j, v, D)$ of the rigid standard Weyl multiplet are detailed in [18], and we simply state the result[29]

$$
\begin{aligned}
v &= \frac{1}{4\tilde{\kappa}} \mathrm{d}\mathcal{Y} \,, \\
D &= 2 \left( \sum_{i=1}^{3} a_i^2 + 2 a_{\text{tot}} - a_{\text{tot}}^2 \right) \tilde{\kappa}^2 \,, \\
V^i{}_j &= -\frac{\mathrm{i}}{2} \Big[ (a_{\text{tot}} - 1)\mathcal{Y} + \mathrm{d}(\phi_1 + \phi_2 + \phi_3) \Big] (\sigma^3)^i{}_j \,,
\end{aligned} \tag{2.21}
$$

where $a_{\text{tot}} = \sum_{i=1}^{3} a_i$. This solution for the background fields in the standard Weyl multiplet satisfies the supersymmetry equations (2.8) with the following (conformal) Killling spinors

$$
\begin{aligned}
\varepsilon^1 &= \frac{\sqrt{\tilde{\kappa}\tilde{\beta}}}{2\sqrt{2}} \begin{pmatrix} -\mathrm{i} \\ \mathrm{i} \\ 1 \\ -1 \end{pmatrix} \,, \qquad
\varepsilon^2 = \frac{\sqrt{\tilde{\kappa}\tilde{\beta}}}{2\sqrt{2}} \begin{pmatrix} -\mathrm{i} \\ -\mathrm{i} \\ -1 \\ -1 \end{pmatrix} \,, \\
\eta^1 &= \left[ \frac{\mathrm{i}}{6} \left( (1-a_{\text{tot}})\tilde{\kappa}^2 - \frac{4}{\tilde{\kappa}\tilde{\beta}} \right) + \frac{1}{6} \partial_a \log \left( \tilde{\kappa}\tilde{\beta}^2 \right) \gamma^a + \frac{1}{3} v_{ab} \gamma^{ab} \right] \varepsilon^1 \,, \\
\eta^2 &= \left[ -\frac{\mathrm{i}}{6} \left( (1-a_{\text{tot}})\tilde{\kappa}^2 - \frac{4}{\tilde{\kappa}\tilde{\beta}} \right) + \frac{1}{6} \partial_a \log \left( \tilde{\kappa}\tilde{\beta}^2 \right) \gamma^a + \frac{1}{3} v_{ab} \gamma^{ab} \right] \varepsilon^2 \,.
\end{aligned} \tag{2.22}
$$

---

[29]There is some residual freedom in our choice of solution, which we have fixed in order to obtain a (somewhat) simple-looking answer.

We have chosen the frame

$$
\begin{aligned}
e^1 &= \frac{1}{y_3\sqrt{1-y_2^2}}\Big[(y_2^2-1)\mathrm{d}y_1 - y_1 y_2\,\mathrm{d}y_2\Big], \\
e^2 &= \frac{y_1 y_3}{\sqrt{1-y_2^2}}\Big[(\mathrm{d}\phi_1 - \mathrm{d}\phi_3) + \frac{a_3 - a_1}{\tilde{\beta}}\mathcal{X}\Big], \\
e^3 &= \frac{1}{\sqrt{1-y_2^2}}\,\mathrm{d}y_2, \\
e^4 &= \frac{y_2}{\sqrt{1-y_2^2}}\Big[-\mathrm{d}\phi_2 + \frac{1+a_2}{\tilde{\beta}}\mathcal{X}\Big], \\
e^5 &= \frac{1}{\tilde{\kappa}\tilde{\beta}}\mathcal{X} + \frac{1}{\tilde{\kappa}}\mathcal{Y},
\end{aligned}
\tag{2.23}
$$

with the definitions

$$
\mathcal{X} = \sum_{i=1}^{3} y_i^2 \mathrm{d}\phi_i, \qquad \mathcal{Y} = \tilde{\kappa}^2 \sum_{i=1}^{3} a_i y_i^2 \mathrm{d}\phi_i, \qquad \tilde{\beta} = 1 + \sum_i a_i y_i^2,
\tag{2.24}
$$

and the 5d gamma matrices are given by

$$
\begin{aligned}
\gamma_1 &= \sigma_3 \otimes \mathbb{1}_{2\times 2}, & \gamma_2 &= \sigma_2 \otimes \mathbb{1}_{2\times 2}, & \gamma_3 &= -\sigma_2 \otimes \sigma_3, \\
\gamma_5 &= -\sigma_2 \otimes \sigma_2, & \gamma_5 &= -\sigma_2 \otimes \sigma_1,
\end{aligned}
\tag{2.25}
$$

with $\sigma_i$ the standard Pauli matrices.

**Flavor multiplet solution**

Next we present the solution for $n_V$ flavor (background) vector multiplets $\mathcal{V}_{\mathrm{f}}^I$, $I = 1,\ldots,n_V$, coupled to the standard Weyl multiplet $\mathcal{SW}$. As mentioned above, we fix the scalar $M_{\mathrm{f}}^I$ to be the constant mass $m_{\mathrm{f}}^I$ of the $I^{\mathrm{th}}$ vector multiplet. Then, we find the following solution to the vector multiplet supersymmetry condition (2.12) (of course this is also a solution to the standard gauge-fixing of Poincaré supergravity given in (2.16))[30]

Flavor solution $\mathcal{V}_{\mathrm{f}}^I$ :

$$
\begin{aligned}
M_{\mathrm{f}}^I &= \mu_{\mathrm{f}}^I, \\
(W_{\mathrm{f}}^I)_\mu \mathrm{d}x^\mu &= \mu_{\mathrm{f}}^I\big(1 - \tilde{\beta}\tilde{\kappa}\big)\Big(\frac{1}{(\tilde{\kappa}\tilde{\beta})^2}\mathcal{X} + \frac{1}{\tilde{\kappa}^2\tilde{\beta}}\mathcal{Y}\Big), \\
(Y_{\mathrm{f}}^I)^i{}_j &= \mathrm{i}\mu_{\mathrm{f}}^I\Big[\frac{1-a_{\mathrm{tot}}}{\tilde{\beta}} + \frac{(a_{\mathrm{tot}}-1)\tilde{\kappa}}{2} + \frac{2}{\tilde{\beta}\tilde{\kappa}} - \frac{3}{(\tilde{\kappa}\tilde{\beta})^2}\Big](\sigma^3)^i{}_j.
\end{aligned}
\tag{2.26}
$$

**KK multiplet solution**

For this solution, we would like a connection between the KK gauge field $W_{\mathrm{KK}}$ in the vector multiplet and the graviphoton in the 6d metric (2.3). This will lead to a *different* solution for the vector multiplet than the one in (2.26), and in particular, the mass $M_{\mathrm{KK}}$ will not be constant. Of course this is expected, since there is a warping factor in the 6d metric. Although the two solutions are distinct, we spell out a way to relate them in Appendix D.

---

[30]Notice, that for our purposes it is crucial that the solution presented here is continuously connected to the round $S^5$ solution, presented in [18]. In particular, this means that $W_{\mathrm{f}}^I$ vanishes in the round limit, $a_i \to 0$ – there is another general class of "topologically non-trivial" solutions, which is related to 5d instanton backgrounds.

Then, the "KK solution" $\mathcal{V}_{\text{KK}}$ is given by

$$
\text{KK solution } \mathcal{V}_{\text{KK}}: \quad
\begin{aligned}
M_{\text{KK}} &= -m_{\text{KK}}\tilde{\kappa}, \\
(W_{\text{KK}})_\mu \mathrm{d}x^\mu &= m_{\text{KK}}\mathcal{Y}, \\
(Y_{\text{KK}})^i{}_j &= \frac{\mathrm{i}m_{\text{KK}}}{2}\tilde{\kappa}^2(1-a_{\text{tot}})(\sigma_3)^i{}_j,
\end{aligned}
\tag{2.27}
$$

where we remark that indeed $W_{\text{KK}}$ precisely agrees with the graviphoton gauge field in (2.3), and we can further pick the constant $m_{\text{KK}} = \frac{2\pi\mathrm{i}}{\beta}$ to fully match to 6d.

**Poincaré solution in the "KK gauge"**

Finally, in order to evaluate the last supersymmetric Chern-Simons term we require a solution to Poicaré supergravity given by imposing the KK gauge-fixing conditions (2.17). This is because the relevant Chern-Simons term, $A \wedge \mathrm{tr}(R \wedge R)$, contains the $U(1)_{KK}$ gauge field, $A$, rather than a flavor gauge field. Then, the corresponding solutions are given by the standard Weyl solution (2.21) together with the KK solution (2.27) – which is now a compensator vector multiplet within the Poincaré multiplet, *i.e.* $\hat{\mathcal{V}} = \mathcal{V}_{\text{KK}}$ – and the following compensator linear multiplet

$$
\begin{aligned}
\hat{L}^i{}_j &= \frac{\mathrm{i}}{2}(\sigma_3)^i{}_j, \\
\hat{E}_\mu \mathrm{d}x^\mu &= \frac{y_1^2 y_3^2}{1-y_2^2}\left(\frac{4}{\tilde{\beta}}(a_1-a_3)+(a_1^2+a_3^2)\tilde{\kappa}^2\right)\left[\frac{(a_1-a_3)}{\tilde{\beta}}\mathcal{X}-\mathrm{d}\phi_1+\mathrm{d}\phi_3\right] \\
&\quad + \frac{y_2^2}{1-y_2^2}\left(\frac{4}{\tilde{\beta}}(1+a_2)+(a_2^2-1)\tilde{\kappa}^2-3\right)\left[\frac{(1+a_2)}{\tilde{\beta}}\mathcal{X}-\mathrm{d}\phi_2\right], \\
\hat{N} &= \left(\frac{4}{\tilde{\kappa}\tilde{\beta}}+(a_{\text{tot}}-1)\tilde{\kappa}\right).
\end{aligned}
\tag{2.28}
$$

This solution satisfies the supersymmetry conditions (2.8), (2.12) and (2.15) together with the KK gauge fixing conditions (2.17), and thus constitutes a solution to KK gauge-fixed Poincaré supergravity.

## 2.4 Supersymmetric Euclidean Chern-Simons terms and their evaluation on the squashed $S^5$

With the squashed $S^5$ solutions for the various supergravity multiplets in hand, we can proceed with evaluating the explicit 5d supersymmetric Chern-Simons terms corresponding to the (minimal) supersymmetric completion of the pieces in the 5d effective theory, arising in the Cardy limit of the 6d superconformal index. We first introduce their explicit expressions for Riemannian manifolds, which requires a careful Wick rotation from the expressions in the literature, and then provide their evaluation on the rigid supersymmetric background.[31]

There are four supersymmetric Chern-Simons terms: FFF, $F_f F_f F$, FWW, and FRR, whose linear combinations correspond to $I_1, \ldots, I_4$ in (1.21).[32] The first two fall into the same class in the classification of [18], while FWW and FRR constitute a subset of the remaining three classes. The reason we can omit the F-Ric² term is that it can be related to a linear combination of FWW and FRR by a field redefinition.

---

[31] In the following, we shall present the Euclidean version of the relevant supersymmetric Chern-Simons terms; this is somewhat subtle and to the knowledge of the authors has not been discussed in the literature before.

[32] Here, we are using the notation of [18], and label the Chern-Simons terms by the featured fields: F and $F_f$ for the KK and flavor-gauge fields/field strengths, W for the Weyl tensor, and R for the Ricci scalar.

The FFF and $F_f F_f F$ terms can be treated uniformly by first assuming a generic set of vector multiplets labeled by $I, J, K \in \{0, 1, \dots, n_V - 1\}$ with structure constant $c_{IJK}$. The (Euclidean) supersymmetry completion of the $A \wedge dA \wedge dA$ term in the rigid limit can be written as follows [19, 20] (see also [89])[33]

$$
\begin{aligned}
S_1(\mathcal{V}_I) = \int_{\mathcal{M}_5} c_{IJK} \Bigg[ &\frac{1}{2} W^I \wedge F^J(W) \wedge F^K(W) - \frac{3}{2} M^I F^J(W) \wedge *F^K(W) \\
&+ \frac{3}{2} M^I dM^J \wedge *dM^K - 3M^I M^J \left(2F^K(W) + M^K \nu\right) \wedge *\nu \\
&+ M^I \left(3(Y^J)_{ij}(Y^K)^{ij} + \frac{1}{4} M^J M^K \left[\frac{R}{2} - D\right]\right) \text{vol}_5 \Bigg],
\end{aligned}
\tag{2.29}
$$

where we denoted by $F^I(W)$ the two-form field strength of the gauge field $W^I$, and the volume five-form is explicitly given by[34]

$$
\text{vol}_5 = \sqrt{g} \, dy_1 \wedge dy_2 \wedge d\phi_1 \wedge d\phi_2 \wedge d\phi_3.
\tag{2.30}
$$

### 2.4.1  FFF **term**

We start by considering the first supersymmetric Chern-Simons term, given by the supersymmetry completion of the 5d Chern-Simons action

$$
A \wedge dA \wedge dA,
\tag{2.31}
$$

where (as before) $A$ is the $U(1)_{KK}$ graviphoton. The minimal supersymmetric extension is simply given by a $U(1)$ vector multiplet $\mathcal{V}_{KK}$ coupled to the standard Weyl multiplet $\mathcal{SW}$. Therefore, there is no need to be working in Poincaré supergravity, and hence, we are not required to impose any gauge-fixing conditions.

Here, we are solely dealing with the gauge fields arising from $U(1)_{KK}$-photons, and thus we may set $I = J = K = 0$ and $c_{000} = 1$ in (2.29). We may then plug in our solutions into the supersymmetric action (2.29), i.e. we take $\mathcal{V} \equiv \mathcal{V}_{KK}$, with corresponding solutions in (2.21) and (2.27).[35] We readily observe that up to some total derivative terms, the integrand reduces to the 5d contact volume, i.e.

$$
S_1(\mathcal{V}_{KK}, \mathcal{V}_{KK}, \mathcal{V}_{KK}) = \frac{m_{KK}^3}{2} \int_{\mathcal{M}_5} \eta \wedge d\eta \wedge d\eta + \int_{\mathcal{M}_5} d * (\cdots),
\tag{2.32}
$$

where $\eta$ is the contact 1-form on the squashed $S^5$, given by

$$
\eta = \mathcal{X} + \tilde{\beta} \mathcal{Y},
\tag{2.33}
$$

where $\mathcal{X}$ and $\mathcal{Y}$ are given in (2.24). This is true for a more general family of backgrounds. Then, a simple application of the Duistermaat-Heckman fixed-point formula [93] shows that

$$
S_1(\mathcal{V}_{KK}, \mathcal{V}_{KK}, \mathcal{V}_{KK}) = \frac{m_{KK}^3}{2} \frac{(2\pi)^3}{\omega_1 \omega_2 \omega_3}.
\tag{2.34}
$$

---

[33]Here, we use the standard conventions for the 5d Hodge star operator, i.e. for $p$-forms $\alpha$ and $\beta$ we have $\alpha \wedge *\beta = \frac{1}{p!} \alpha_{\mu_1 \cdots \mu_p} \beta^{\mu_1 \cdots \mu_p} * 1$.

[34]This action also gives rise to the third supersymmetric Chern-Simons term $S_3$ by replacing two instances of the vector multiplet by composite expressions in terms of the linear multiplet, i.e. we have $S_1(\hat{\mathcal{V}}, \underline{\mathcal{V}}, \underline{\mathcal{V}})$, where by $\underline{\mathcal{V}}$ we denote the vector multiplet expressed in terms of a linear compensator multiplet; See below.

[35]One could also simply evaluate the standard Poincaré supergravity action, which includes a compensator linear multiplet piece. This would not correspond to the minimally extended version of the corresponding Chern-Simons term, and it would only contribute to the subleading pieces $\mathcal{O}(\beta^0)$ in the $\beta$-expansion.

### 2.4.2 $F_f F_f F$ term

We now turn towards the supersymmetric completion of the term

$$A \wedge \text{Tr}\left(F_{G_f} \wedge F_{G_f}\right), \tag{2.35}$$

where we recall that $F_{G_f}$ is the field strength for a background flavor multiplet. For simplicity, we focus on the background

$$F_{G_f} = F_f^I H_f^I = \mathrm{d}W_I^f H_f^I, \tag{2.36}$$

where $H_f^I$ are the generators of the Cartan subalgebra of the Lie algebra of the flavor symmetry $G_f$ with the normalization $\text{Tr}(H_f^I H_f^J) = \delta^{IJ}$.

We start again with the action in equation (2.29), where we now pick two of the vector multiplets to be pure flavor multiplets,

$$\mathcal{V}_f^I = \left(\left(W_f^I\right)_\mu, M_f^I, \left(\Omega_f^I\right)_\alpha^i, \left(Y_f^I\right)^{ij}\right), \tag{2.37}$$

while the third one is given by the KK vector multiplet, $\mathcal{V}_{KK}$. The resulting action, $S_1(\mathcal{V}_f^I, \mathcal{V}_f^J, \mathcal{V}_{KK})$ with $c_{IJK} = \delta_{(IJ}\delta_{K)0}$, is then given by

$$
\begin{aligned}
S_1\left(\mathcal{V}_f^I, \mathcal{V}_f^J, \mathcal{V}_{KK}\right) = \int_{\mathcal{M}_5} \delta_{IJ} \Bigg[ &\frac{1}{6}\left(2W_f^I \wedge F_{KK} + W_{KK} \wedge F_f^I\right) \wedge F_f^J \\
&- \frac{1}{2}\left(2M_f^I F_{KK} + M_{KK}F_f^I\right) \wedge *F_f^J + \frac{1}{2}\left(2M_f^I \mathrm{d}M_{KK} + M_{KK}\mathrm{d}M_f^I\right) \wedge *\mathrm{d}M_f^J \\
&- 2M_f^I\left(2M_{KK}F_f^J + M_f^J F_{KK}\right) \wedge *\nu - 3M_f^I M_f^J M_{KK}\nu \wedge *\nu \\
&+ (Y_f^I)_{ij}\left(2M_f^J(Y_{KK})_{ij} + M_{KK}(Y_f^J)_{ij}\right)\text{vol}_5 - \frac{1}{4}M_f^I M_f^J M_{KK}\left(D - \frac{R}{2}\right)\text{vol}_5 \Bigg],
\end{aligned} \tag{2.38}
$$

and we plug in the standard Weyl solution (2.21) together with one instance of the KK solution (2.27) and two instances of the flavor solution (2.26). By the same arguments as before, we end up with

$$S_1(\mathcal{V}_f^I, \mathcal{V}_f^J, \mathcal{V}_{KK}) = \frac{m_{KK}}{2}\frac{(2\pi)^3}{\omega_1\omega_2\omega_3}\mu_f^2, \tag{2.39}$$

where $\mu_f^2 = \sum_I (\mu_f^I)^2$.

### 2.4.3 FWW term

Let us now turn to the remaining two supersymmetric Chern-Simons terms. They are given by the supersymmetry completion of the terms $A \wedge \text{tr}(R \wedge R)$ and $A \wedge \text{Tr}(F_R \wedge F_R)$, where (as before) $A$ is the $U(1)_{KK}$-photon and $F_R = F(V)$ the background $\mathfrak{su}(2)_R$ field strength. These terms decompose into (a linear combination of) two supersymmetric higher-derivative terms, which are themselves the supersymmetric completion of the Weyl- and Ricci-squared higher-derivative actions in 5d. We shall first focus on the flavor-Weyl$^2$ (FWW) term, and below discuss the flavor-Ricci$^2$ (FRR) one.

First, we turn to the supersymmetric completion of the Weyl-squared higher-derivative term. This term gives the supersymmetric completions of a combination of the following Chern-Simons terms

$$A \wedge \text{tr}(R \wedge R) \quad \text{and} \quad A \wedge \text{Tr}(F_R \wedge F_R). \tag{2.40}$$

As in the case of the FFF supersymmetric Chern-Simons term, the FWW term can be written purely in terms of the vector multiplet coupled to the standard Weyl multiplet. It was first

written down in [21] (see also [89]), and its Euclidean (Wick rotated) version in the rigid limit is given as follows[36]

$$
\begin{aligned}
S_2(\mathcal{V}_I) = \int_{\mathcal{M}_5} c_I \sqrt{g} \Bigg[ & M^I \left( \frac{1}{8} C^2 - (v^2)^2 + \frac{1}{12} D^2 \right) + \left( \frac{D}{6} - \frac{4}{9} v^2 \right) v_{\mu\nu} F^I(W)^{\mu\nu} \\
& + C_{\mu\nu\rho\sigma} v^{\mu\nu} \left( \frac{1}{3} M^I v^{\rho\sigma} + \frac{1}{2} F^I(W)^{\rho\sigma} \right) - \frac{4}{3} (Y^I)_{ij} F(V)_{\mu\nu}{}^{ij} v^{\mu\nu} \\
& - \frac{1}{12} \epsilon^{\mu\nu\rho\sigma\lambda} (W^I)_\mu \left( \frac{3}{4} C_{\nu\rho\tau\delta} C_{\sigma\lambda}{}^{\tau\delta} - F(V)_{\nu\rho}{}^{ij} F(V)_{\sigma\lambda ij} \right) \\
& - \epsilon_{\mu\nu\rho\sigma\lambda} F^I(W)^{\mu\nu} \left( \frac{2}{3} v^{\rho\tau} \nabla_\tau v^{\sigma\lambda} + v^\rho{}_\tau \nabla^\sigma v^{\lambda\tau} \right) \\
& - \frac{1}{3} M^I \left( F(V)_{\mu\nu}{}^{ij} F(V)^{\mu\nu}{}_{ij} + 4\nabla_\nu v_{\mu\rho} \nabla^\mu v^{\nu\rho} - 8 v_{\mu\nu} \nabla^\nu \nabla_\rho v^{\mu\rho} \right) \\
& + \frac{1}{9} M^I \left( 16 R^{\nu\rho} v_{\mu\nu} v^\mu{}_\rho + 12 \nabla_\mu v_{\nu\rho} \nabla^\mu v^{\nu\rho} - 2R v^2 \right) \\
& + \frac{1}{3} F^I(W)^{\mu\nu} \left( \frac{1}{3} v_{\mu\nu} v^2 + 4 v_{\mu\rho} v^{\rho\lambda} v_{\nu\lambda} \right) + 4 M^I v^{\mu\nu} v_{\nu\rho} v^{\rho\sigma} v_{\sigma\mu} \\
& + \frac{2}{3} M^I \epsilon_{\mu\nu\rho\sigma\lambda} v^{\mu\nu} v^{\rho\sigma} \nabla_\tau v^{\lambda\tau} \Bigg],
\end{aligned}
\tag{2.41}
$$

where we have defined

$$
C^2 = C^{\mu\nu\rho\sigma} C_{\mu\nu\rho\sigma}, \tag{2.42}
$$

with the Weyl tensor given by

$$
\begin{aligned}
C_{\mu\nu\rho\sigma} = & R_{\mu\nu\rho\sigma} - \tfrac{1}{3}(g_{\mu\rho} R_{\nu\sigma} - g_{\nu\rho} R_{\mu\sigma} - g_{\mu\sigma} R_{\nu\rho} + g_{\nu\sigma} R_{\mu\rho}) \\
& + \tfrac{1}{12}(g_{\mu\rho} g_{\nu\sigma} - g_{\mu\sigma} g_{\nu\rho}) R.
\end{aligned}
$$

As before, we use the solution for the KK vector multiplet $\mathcal{V}_{KK}$ (2.27) together with the standard Weyl solution (2.21), and a straightforward yet tedious computation yields

$$
\begin{aligned}
S_2(\mathcal{V}_{KK}) &= \frac{(a_1^2 + a_2^2 + a_3^2) - (a_1 a_2 + a_1 a_3 + a_2 a_3)}{3\omega_1 \omega_2 \omega_3} (2\pi)^3 m_{KK} \\
&= \left[ \frac{\omega_1^2 + \omega_2^2 + \omega_3^2}{2\omega_1 \omega_2 \omega_3} - \frac{(\omega_1 + \omega_2 + \omega_3)^2}{6\omega_1 \omega_2 \omega_3} \right] (2\pi)^3 m_{KK},
\end{aligned}
\tag{2.43}
$$

where we have set $c_I = \delta_{I,0}$ indicating that we are dealing with a single Abelian $U(1)_{KK}$ gauge field.

### 2.4.4 FRR term

Finally, the FRR supersymmetric Chern-Simons term coincides precisely with the supersymmetric completion of the piece

$$
A \wedge \mathrm{Tr}(F_R \wedge F_R). \tag{2.44}
$$

It can be written in terms of Poincaré supergravity, which we introduced as a gauge-fixing of the standard Weyl multiplet above. Notice that this term explicitly breaks conformal invariance.

The full Lagrangian was first constructed in [89] (in Lorentzian signature), by taking the vector multiplet action (2.29), and then writing part of the vector multiplet fields as composite

---

[36]Here, we mostly follow the notation in [89], but with $D_{here} = 16 D_{there} + \frac{128}{3} T_{there}^2$, $v_{here} = 4 T_{there}$, as well as $M^I_{here} = -\rho^I_{there}$.

expressions in terms of the gauge-fixed linear multiplet. Therefore, we require two steps to get the Lagrangian; we write down the vector multiplet action and then replace the underlined fields by their composite expressions, which we provide further below. The Ricci scalar squared term then arises from the $\underline{Y}_{ij}\underline{Y}^{ij}$ term. By choosing $c_{I,0,0} = c_I$ and all other $c_{IJK}$ to zero in the vector multiplet action, we obtain the following (Euclidean) Ricci scalar squared action in the rigid limit

$$
\begin{aligned}
S_3(\hat{\mathcal{V}}_I) = \int_{\mathcal{M}_5} c_I \sqrt{g} \Bigg[ & \hat{M}^I \underline{Y}_{ij}\underline{Y}^{ij} - 2\underline{\rho}\,\underline{Y}^{ij}(\hat{Y}^I)_{ij} + \frac{1}{8}\hat{M}^I \underline{\rho}^2 R - \frac{1}{4}\hat{M}^I \underline{F}_{\mu\nu}\underline{F}^{\mu\nu} \\
& + \frac{1}{2}\underline{\rho}\,\underline{F}^{\mu\nu}\hat{F}^I(\hat{W})_{\mu\nu} - \frac{1}{2}\hat{M}^I \partial_\mu\underline{\rho}\,\partial^\mu\underline{\rho} - \hat{M}^I \underline{\rho}\,\partial^\mu\partial_\mu\underline{\rho} \\
& - \frac{1}{4}\hat{M}^I \underline{\rho}^2 \left(D + 6v^2\right) - \underline{\rho}^2 \hat{F}^I(\hat{W})_{\mu\nu}v^{\mu\nu} + 2\hat{M}^I \underline{\rho}\,\underline{F}_{\mu\nu}v^{\mu\nu} \\
& + \frac{1}{8}\epsilon_{\mu\nu\rho\sigma\lambda}(\hat{W}^I)^\mu \underline{F}^{\nu\rho}\underline{F}^{\sigma\lambda} \Bigg],
\end{aligned}
\tag{2.45}
$$

where the composite expressions (in the rigid limit) are given by

$$
\begin{aligned}
\underline{\rho} &= \frac{\hat{N}}{(\hat{L}^{ij}\hat{L}_{ij})^{\frac{1}{2}}}, \\
\underline{F}_{\mu\nu} &= 2\nabla_{[\mu}((\hat{L}^{ij}\hat{L}_{ij})^{-\frac{1}{2}}\hat{E}_{\nu]}) + \frac{2}{(\hat{L}^{ij}\hat{L}_{ij})^{\frac{1}{2}}}F(V)_{\mu\nu}{}^{ij}\hat{L}_{ij} - \frac{2}{(\hat{L}^{ij}\hat{L}_{ij})^{\frac{3}{2}}}\hat{L}^l_k D_{[\mu}\hat{L}^{kp}D_{\nu]}\hat{L}_{lp}, \\
\underline{Y}_{ij} &= \frac{3}{8(\hat{L}^{ij}\hat{L}_{ij})^{\frac{1}{2}}}\hat{L}_{ij}R - \frac{1}{(\hat{L}^{ij}\hat{L}_{ij})^{\frac{1}{2}}}\partial^\mu D_\mu\hat{L}_{ij} - \frac{2}{(\hat{L}^{ij}\hat{L}_{ij})^{\frac{1}{2}}}V_\mu{}^i{}_k D^\mu L^{jk} \\
& + \frac{1}{(\hat{L}^{ij}\hat{L}_{ij})^{\frac{3}{2}}}D_a\hat{L}_{k(i}D^a\hat{L}_{j)m}\hat{L}^{km} - \frac{1}{4(\hat{L}^{ij}\hat{L}_{ij})^{\frac{3}{2}}}\hat{N}^2\hat{L}_{ij} \\
& + \frac{1}{4(\hat{L}^{ij}\hat{L}_{ij})^{\frac{3}{2}}}\hat{E}_\mu\hat{E}^\mu\hat{L}_{ij} + \frac{1}{4(\hat{L}^{ij}\hat{L}_{ij})^{\frac{1}{2}}}\left(D - 2v^2\right)\hat{L}_{ij} - \frac{1}{(\hat{L}^{ij}\hat{L}_{ij})^{\frac{3}{2}}}\hat{E}_\mu\hat{L}_{k(i}D^\mu\hat{L}_{j)}{}^k,
\end{aligned}
\tag{2.46}
$$

with $v^2$ defined as

$$
v^2 = v^{\mu\nu}v_{\mu\nu},
\tag{2.47}
$$

the covariant derivative $D_\mu L^{ij}$ given by

$$
D_\mu\hat{L}^{ij} = \partial_\mu\hat{L}^{ij} + 2V_\mu{}^{(i}{}_j\hat{L}^{j)k},
\tag{2.48}
$$

and the symmetrization defined as as $X_{(i}Y_{j)} = \frac{1}{2}(X_iY_j + X_jY_i)$, and similarly for the antisymmetrization.

We can now plug in our explicit supersymmetric solutions for KK gauge-fixed Poincaré supergravity, *i.e.* equations (2.8), (2.12) and (2.15), with gauge fixing conditions (2.17). Again, a tedious analysis shows that it integrates to

$$
S_3(\hat{\mathcal{V}}) = -\frac{(\omega_1 + \omega_2 + \omega_3)^2}{\omega_1\omega_2\omega_2}(2\pi)^3 m_{\text{KK}},
\tag{2.49}
$$

where we have set $c_I = \delta_{I,0}$ as we are to dealing with a single Abelian $U(1)_{\text{KK}}$-photon.

## 2.5 Summary

Let us now conclude this technical section with a summary of our results. By carefully evaluating the supersymmetric completion of the four Chern-Simons terms appearing in the 5d

effective action, and imposing the following relation between the KK-masses $m_{\mathrm{KK}}$ to the $S^1$ circumference $\beta$

$$m_{\mathrm{KK}} = \frac{2\pi\mathrm{i}}{\beta}, \tag{2.50}$$

we have shown that

$$
\begin{aligned}
I_1 &\equiv 2S_1\left(\mathcal{V}_{\mathrm{KK}}, \mathcal{V}_{\mathrm{KK}}, \mathcal{V}_{\mathrm{KK}}\right) = \int_{\mathcal{M}_5} A \wedge \mathrm{d}A \wedge \mathrm{d}A + \text{ SUSY completion} \\
&= \frac{(2\pi)^3}{\omega_1\omega_2\omega_3}\left(\frac{2\pi\mathrm{i}}{\beta}\right)^3, \\
I_2 &\equiv -4\left[S_2\left(\mathcal{V}_{\mathrm{KK}}\right) - \frac{1}{6}S_3(\hat{\mathcal{V}})\right] = \int_{\mathcal{M}_5} A \wedge \mathrm{tr}\left(R \wedge R\right) + \text{ SUSY completion} \\
&= -2(2\pi)^3 \frac{(\omega_1^2 + \omega_2^2 + \omega_3^2)}{\omega_1\omega_2\omega_3}\left(\frac{2\pi\mathrm{i}}{\beta}\right), \\
I_3 &\equiv -\frac{1}{2}S_3(\hat{\mathcal{V}}) = \int_{\mathcal{M}_5} A \wedge \mathrm{Tr}\left(F_R \wedge F_R\right) + \text{ SUSY completion} \\
&= \frac{(2\pi)^3}{2}\frac{(\omega_1 + \omega_2 + \omega_3)^2}{\omega_1\omega_2\omega_3}\left(\frac{2\pi\mathrm{i}}{\beta}\right), \\
I_4 &\equiv 2S_1\left(\mathcal{V}_{\mathrm{KK}}, \mathcal{V}_{\mathrm{f}}^I, \mathcal{V}_{\mathrm{f}}^J\right) = \int_{\mathcal{M}_5} A \wedge \mathrm{Tr}\left(F_{G_{\mathrm{f}}} \wedge F_{G_{\mathrm{f}}}\right) + \text{ SUSY completion} \\
&= \frac{(2\pi)^3}{\omega_1\omega_2\omega_3}\mu_{\mathrm{f}}^2\left(\frac{2\pi\mathrm{i}}{\beta}\right).
\end{aligned}
\tag{2.51}
$$

## 3 Deriving the Chern-Simons levels

Having shown that the 5d supersymmetric Chern-Simons terms produce the form of the Cardy formula, it remains to determine the Chern-Simons levels $\kappa_i$ in (1.14). Let us revisit the origin of these terms from the 6d perspective: they come from integrating out *massive* Kaluza-Klein (KK) modes, and are accompanied by non-local pieces arising from the massless degrees of freedom. The latter are *uncharged* under the $U(1)_{\mathrm{KK}}$ symmetry, so their contributions are of order $\mathcal{O}(\beta^0)$ or $\mathcal{O}(\log\beta)$. Since we are interested in the parts of the 5d effective action that are singular as $\beta \to 0$, we only need to take into account the massive KK contributions, and doing so allows us to determine the Chern-Simons levels. We emphasize that such 5d Chern-Simons terms are unaffected by the local counter-terms in 6d.

The Chern-Simons terms that arise from integrating out massive KK modes can be computed explicitly for various 6d free fields. For interacting SCFTs, our strategy is to connect them to free theories by renormalization group flows on the vacuum moduli space. To see that this produces the correct answer at the origin (*i.e.* the interacting superconformal point), we recall the argument given in Section 1.4 that in the 5d *Wilsonian* effective action, any dependence $\kappa(\phi)$ on the moduli fields violates background gauge invariance.

For Higgs branch flows, the free field computation allows us to determine the precise relation between the Chern-Simons levels and perturbative anomaly coefficients. This completes the proof of the 6d Cardy formula for general SCFTs with a Higgs branch. We also carry out the computation for tensor branch flows, and discover that additional contributions from the BPS strings wrapping the $S^1$, are essential, though to compute them we need further information about such objects.

## 3.1 Integrating out free field KK modes

The 5d Chern-Simons terms arise from integrating out KK modes from the $S^1$ reduction of 6d tensor, vector and hypermultiplets. It is well-known that only chiral fermions $\psi_\mp$ and (anti)-self-dual 2-forms $B$ can generate such Chern-Simons terms. The Chern-Simons terms of the first two types in (1.21) are given in [94]. The result is one-loop exact due to the usual non-renormalization argument. For a conjugate pair of KK modes of $U(1)_{KK}$ charge $\pm n$, we get the following contributions

$$
\begin{aligned}
\mathcal{I}^n_{\psi_-} &= \frac{1}{48\pi^2}n^3 I_1 + \frac{1}{384\pi^2}n I_2, \\
\mathcal{I}^n_{\psi_+} &= -\frac{1}{48\pi^2}n^3 I_1 - \frac{1}{384\pi^2}n I_2, \\
\mathcal{I}^n_B &= -\frac{4}{48\pi^2}n^3 I_1 + \frac{8}{384\pi^2}n I_2,
\end{aligned}
\tag{3.1}
$$

where the various Chern-Simons terms are given in (1.21).

For $2m$ fermions transforming in the $\mathbf{2m}$ representation of $USp(2m)$, a similar computation gives

$$
\begin{aligned}
\tilde{\mathcal{I}}^n_{\psi_\mp} &= \pm\frac{1}{32\pi^2}n\widetilde{I}(USp(2m)), \\
\widetilde{I}(USp(2m)) &= \int A \wedge \mathrm{Tr}\left(F_{USp(2m)} \wedge F_{USp(2m)}\right) + \text{SUSY completion}.
\end{aligned}
\tag{3.2}
$$

A tensor multiplet contains the fields $(B, \psi^i_-)$, while a vector multiplet only contains the fermions $\psi^i_+$, where $i \in \{1, 2\}$ is the fundamental index for $SU(2)_R$. On the other hand, $n_H$ hypermultiplets contain $2n_H$ fermions $\psi^i_-$, which are uncharged under $SU(2)_R$ and transformed in $\mathbf{2n_H}$ representation of $USp(2n_H)$.

Putting these contributions together and using zeta function regularization to perform the sum over KK modes,

$$
\sum_{n=1}^{\infty}(n^3) = \frac{1}{120}, \qquad \sum_{n=1}^{\infty}(n) = -\frac{1}{12},
\tag{3.3}
$$

we obtain

$$
\begin{aligned}
\mathcal{I}_T &= \frac{2-4}{48\pi^2}\frac{1}{120}I_1 - \frac{2+8}{384\pi^2}\frac{1}{12}I_2 - \frac{2}{32\pi^2}\frac{1}{12}I_3, \\
\mathcal{I}_V &= \frac{-2}{48\pi^2}\frac{1}{120}I_1 - \frac{-2}{384\pi^2}\frac{1}{12}I_2 - \frac{-2}{32\pi^2}\frac{1}{12}I_3, \\
\mathcal{I}^{n_H}_H &= \frac{2}{48\pi^2}\frac{1}{120}I_1 - \frac{2}{384\pi^2}\frac{1}{12}I_2 - \frac{2}{32\pi^2}\frac{1}{12}I^{USp(2n_H)}_4,
\end{aligned}
\tag{3.4}
$$

where $I_3$ and $I^{USp(2n_H)}_4$ are

$$
I_3 = \widetilde{I}(SU(2)_R \cong USp(2)), \qquad I^{USp(2n_H)}_4 = \widetilde{I}(USp(2n_H)).
\tag{3.5}
$$

Thus, we conclude that

$$
\begin{aligned}
&n_T\mathcal{I}_T + n_V\mathcal{I}_V + n_H\mathcal{I}_H \\
&= \frac{(n_H - n_V - n_T)}{24\pi^2}\frac{1}{120}I_1 - \frac{(n_H - n_V + 5n_T)}{192\pi^2}\frac{1}{12}I_2 - \frac{(-n_V + n_T)}{16\pi^2}\frac{1}{24}I_3 - \frac{1}{16\pi^2}\frac{1}{24}I^{USp(2n_H)}_4,
\end{aligned}
\tag{3.6}
$$

and comparing with the expression of (1.14) for free fields rewritten as

$$
\begin{aligned}
\mathrm{i}W &= \frac{\mathrm{i}}{8\pi^2}\left(\frac{n_T + n_V - n_H}{360}I_1 + \frac{n_H + 5n_T - n_V}{288}I_2 + \frac{n_T - n_V}{24}I_3 + \frac{1}{24}I^{USp(2n_H)}_4\right) \\
&\quad + \mathcal{O}(\beta^0, \log\beta).
\end{aligned}
\tag{3.7}
$$

## 3.2 On the Higgs branch

On the Higgs branch, the free-field analysis above is sufficient to fully capture the 6d Cardy formula. The reason is as follows. From the perspective of the 6d effective action (see Figure 1), the free-field analysis can potentially miss contributions from extended BPS objects. In Appendix B.2, we classify such objects by studying the central extensions of the supersymmetry algebra, and find that the only extended BPS objects allowed on the Higgs branch are codimension-two vortex branes. However, such objects wrapped on $S^1$ have infinite energy (unlike the BPS strings on the tensor branch), and thus cannot contribute to the Chern-Simons levels in the 5d effective action. We now proceed with explicitly deriving the relation between the Chern-Simons levels $\kappa_i$ in the 5d effective action (1.20) and the (perturbative) anomaly coefficients in the anomaly polynomial (1.15).

Let us assume that the 6d SCFT has a non-Abelian flavor symmetry group $G_f$ and a Higgs branch, whose infrared limit consists of free vector and hypermultiplets. At an arbitrary point on the Higgs branch, the global symmetry group $SU(2)_R \times G_f$ is broken to

$$SU(2)_R \times G_f \; \to \; SU(2)_D \times G_1 \times \cdots \times G_n \,, \tag{3.8}$$

where $SU(2)_D$ is the diagonal subgroup of $SU(2)_R$ and $SU(2)_X$, with $SU(2)_X \times G_1 \times \cdots \times G_n \subset G_f$.[37] The infrared theory consists of $m$ half-hypermultiplets in the doublet of $SU(2)_D$, and free half-hypermultiplets in a pseudo-real representation $\mathbf{r}_1 \oplus \mathbf{r}_2 \oplus \cdots \oplus \mathbf{r}_n$ of $G_1 \times \cdots \times G_n$. Let us denote the dimension of the representation $\mathbf{r}_i$ by $d_{\mathbf{r}_i}$, and $d_{\mathrm{tot}} = \sum_{i=1}^n d_{\mathbf{r}_i}$.

By using the perturbative anomaly matching [64] and global anomaly matching on the Higgs branch, we prove the formula (1.16) for the class of theories considered here. Let us start with the anomaly polynomial of the UV SCFT, which contains the terms

$$I_{UV}^{(8)} \supset \frac{1}{4!} \left[ \delta p_2 + \gamma p_1^2 + \beta c_2(SU(2)_R) p_1 + 24 \mu c_2(G_f) p_1 \right]. \tag{3.9}$$

Under the general symmetry breaking pattern (3.8), we have the following relations involving the second Chern classes:

$$\begin{aligned} c_2(G_f) &= c_2(SU(2)_D) + \sum_{i=1}^n a_i c_2(G_i), \\ c_2(SU(2)_R) &= c_2(SU(2)_D), \end{aligned} \tag{3.10}$$

where $a_i$ are some coefficients that could be determined by the details of the symmetry breaking pattern (3.8). Thus, the UV anomaly polynomial (3.9) admits a rewriting in terms of the residual symmetry group as

$$I_{UV}^{(8)} \supset \frac{1}{4!} \left[ \delta p_2 + \gamma p_1^2 + (\beta + 24\mu) c_2(SU(2)_D) p_1 + 24\mu \left( \sum_{i=1}^n a_i c_2(G_i) \right) p_1 \right]. \tag{3.11}$$

On the Higgs branch, the anomaly polynomial of the IR effective theory receives contributions from the hypermultiplets and contains the terms

$$\begin{aligned} I_{\mathrm{HB}}^{(8)} = \frac{1}{4!} \Bigg\{ &-\frac{1}{60}\left(m + \frac{d_{\mathrm{tot}}}{2}\right) p_2 + \frac{7}{240}\left(m + \frac{d_{\mathrm{tot}}}{2}\right) p_1^2 \\ &+ \frac{m}{2} c_2(SU(2)_D) p_1 + \frac{1}{2} \sum_{i=1}^n T^{G_i}(\mathbf{r}_i) c_2(G_i) p_1 \Bigg\}, \end{aligned} \tag{3.12}$$

---

[37] We have assumed that $G_i$ are simple Lie groups. Our argument can straightforwardly be generalized to the case with $U(1)$ factors appearing on the right-hand side of (3.8).

where $T^G(\mathbf{r})$ is defined by

$$\mathrm{tr}_{\mathbf{r}}(F_G^2) = T^G(\mathbf{r})\mathrm{Tr}(F_G^2). \tag{3.13}$$

We have also used

$$\mathrm{Tr}(F_{\mathfrak{usp}(d_{\mathrm{tot}})}^2) = \mathrm{tr}_{\mathbf{fund}}(F_{\mathfrak{usp}(d_{\mathrm{tot}})}^2) = \sum_i \mathrm{tr}_{\mathbf{r}_i}(F_{G_i}^2) = \sum_i T^{G_i}(\mathbf{r}_i)\mathrm{Tr}(F_{G_i}^2). \tag{3.14}$$

By perturbative anomaly matching [64], we conclude that

$$\delta = -\frac{1}{60}\left(m + \frac{d_{\mathrm{tot}}}{2}\right), \qquad\qquad \gamma = \frac{7}{240}\left(m + \frac{d_{\mathrm{tot}}}{2}\right),$$
$$\beta + 24\mu = \frac{m}{2}, \qquad\qquad 48\mu a_i = T^{G_i}(\mathbf{r}_i). \tag{3.15}$$

Notice that for theories with a Higgs branch,

$$\gamma = -\frac{7}{4}\delta. \tag{3.16}$$

Now, let us turn to the corresponding effective action on the Higgs branch. We start with the general expression

$$\mathrm{i}W_{UV} = \frac{\mathrm{i}\kappa_1}{2880\pi^2}\int_{\mathcal{M}_5} A \wedge \mathrm{d}A \wedge \mathrm{d}A - \frac{\mathrm{i}}{96}\int_{\mathcal{M}_5} A \wedge \left[\frac{2}{3}\left(\kappa_2 - \frac{3}{2}\kappa_3\right)p_1 + 4\kappa_3 c_2^{\mathrm{SU(2)}_R} + 4\kappa_f c_2^G\right], \tag{3.17}$$

which under the Higgs branch symmetry breaking pattern (3.8) reduces to the following expression

$$\mathrm{i}W_{UV} = \frac{\mathrm{i}\kappa_1}{2880\pi^2}\int_{\mathcal{M}_5} A \wedge \mathrm{d}A \wedge \mathrm{d}A$$
$$- \frac{\mathrm{i}}{96}\int_{\mathcal{M}_5} A \wedge \left[\frac{2}{3}\left(\kappa_2 - \frac{3}{2}\kappa_3\right)p_1 + 4(\kappa_3 + \kappa_f)c_2(\mathrm{SU(2)}_D) + 4\kappa_f\left(\sum_{i=1}^n a_i c_2(G_i)\right)\right]. \tag{3.18}$$

On the other hand, plugging in the field content on the Higgs branch to the effective action (3.7), we find

$$W_{\mathrm{HB}} = -\mathrm{i}\frac{m + \frac{d_{\mathrm{tot}}}{2}}{2880\pi^2}\int_{\mathcal{M}_5} A \wedge \mathrm{d}A \wedge \mathrm{d}A$$
$$- \frac{\mathrm{i}}{96}\int_{\mathcal{M}_5} A \wedge \left[\frac{1}{3}\left(m + \frac{d_{\mathrm{tot}}}{2}\right)p_1 - 4m c_2(\mathrm{SU(2)}_D) - 4\sum_{i=1}^n T^{G_i}(\mathbf{r}_i)c_2(G_i)\right]. \tag{3.19}$$

Thus, comparing (3.18) with (3.19), we find the relations

$$\kappa_1 = -\frac{1}{2}(2m + d_{\mathrm{tot}}), \qquad \kappa_2 - \frac{3}{2}\kappa_3 = \frac{1}{4}(2m + d_{\mathrm{tot}}),$$
$$\kappa_3 + \kappa_f = -m, \qquad\qquad \kappa_f a_i = -T^{G_i}(\mathbf{r}_i). \tag{3.20}$$

Hence, from (3.15) and (3.20) we derive the universal formulae for SCFTs with a Higgs branch (with the relation between $\gamma$ and $\delta$ (3.16)):

$$\kappa_1 = 60\delta,$$
$$\kappa_2 - \frac{3}{2}\kappa_3 = -30\delta,$$
$$\kappa_3 = -2\beta,$$
$$\kappa_f = -48\mu. \tag{3.21}$$

This is consistent with the general formulae conjectured in [1], namely

$$
\begin{aligned}
\kappa_1 &= -40\gamma - 10\delta\,, \\
\kappa_2 - \frac{3}{2}\kappa_3 &= 16\gamma - 2\delta\,, \\
\kappa_3 &= -2\beta\,, \\
\kappa_f &= -48\mu\,.
\end{aligned}
\tag{3.22}
$$

### 3.3 On the tensor branch

To extend the analysis to theories without a Higgs branch, a natural step is to derive the coefficients $\kappa_i$ and $\kappa_f$ on the tensor branch. In this section, we compute the "naive" Chern-Simons levels $\kappa_i$ and $\kappa_f$ from the one-loop free field contributions on the tensor branch of several theories, and show that additional contributions must be present. In particular, in theories that also have a Higgs branch, we find a "mismatch" between the two results.[38] Although in 6d there are no massive particles, there are massive (tensionful) BPS strings – including "M-strings" in $\mathcal{N} = (2,0)$ theories and "E-strings" in the E-string theory – that can potentially contribute to the Chern-Simons levels. A key difference between the Higgs branch and the tensor or mixed branch is that BPS strings are absent in the former, but are in general present in the latter. In the M-theory picture, such BPS strings are M2-branes stretched between M5-M5 or M5-M9 branes, and are massive only when there are separations between the M5/M9 branes (*i.e.* on the tensor or mixed branch).

Using supersymmetric localization, the superconformal index can be computed by a contour integral on the complexified tensor branch, and one can explicitly see that the BPS strings crucially contribute to the Cardy limit of the superconformal index. The integrand is a product of three copies of the supersymmetric partition function on $\mathbb{R}^4 \times T^2$ [74,76,79], which can further be written as a product of the partition function of the free tensor multiplets and a sum over the elliptic genera of multiple BPS strings winding around the $T^2$ [95,96]. The Cardy ($\beta \to 0$) limit of the superconformal index reduces to the Cardy limit of the elliptic genera, which is controlled by the central charge (1.1) of the theory living on the BPS strings. Therefore, a potential way to compute the Chern-Simons levels on the tensor branch is to relate them to the 2d central charges of the BPS strings using the above relations. We leave this for future investigation.

Below, we elaborate on the one-loop free field contributions and highlight the difference compared to (1.14) which should come from BPS strings.

In the fourth columns of Table 2 and 3, we give the values of the Chern-Simons levels $\kappa_i$'s and $\kappa_f$ inferred from the free field content of various theories on their tensor branches. The values of the $\kappa_i$'s can be straightforwardly computed using the numbers of the free multiplets given in Table 1. In the following, we give more details on the computation of the Chern-Simons level $\kappa_f$, which only receives contributions from the hypermultiplets. On the tensor branch, the hypermultiplets transform under the $USp(2n_H)$ symmetry. By the one-loop computation in Section 3.1, we find

$$
\kappa_f^{USp(2n_H)} = -1\,.
\tag{3.23}
$$

---

[38]A quick way to see that additional contributions must be present is the following. Many 6d SCFTs have a one-dimensional tensor branch, on which the massless free field content is the same, but have very different Cardy limits according to (1.14). It is plausible that the difference comes from the Green-Schwarz type terms, which appear upon integrating out the appropriate BPS objects appearing on the tensor branch (see Appendix B.2). Likewise, for 4d SCFTs that have a complex one-dimensional Coulomb branch, the contributions from the supersymmetric Wess-Zumino terms – again upon integrating out the heavy BPS states – may be crucial for recovering the Cardy formula at the fixed point.

Table 1: Field content on the tensor branch of various theories.

| Theory | $n_H$ | $n_V$ | $n_T$ |
|---|---|---|---|
| E-string | $N-1$ | 0 | $N$ |
| type-$\mathfrak{g}$ $(2,0)$ | $r_\mathfrak{g}$ | 0 | $r_\mathfrak{g}$ |
| $\mathcal{T}_{N,\Gamma_{A_{k-1}}}$ | $k^2 N$ | $(k^2-1)(N-1)$ | $N-1$ |
| $\mathcal{T}_{N,\Gamma_{D_k}}$ | $2(2k-8)N$ | $k(2k-1)(N-1)+(k-4)(2k-7)N$ | $2N-1$ |
| $\mathcal{T}_{N,\Gamma_{E_6}}$ | 0 | $86N-78$ | $4N-1$ |
| $\mathcal{T}_{N,\Gamma_{E_7}}$ | $16N$ | $160N-133$ | $6N-1$ |
| $\mathcal{T}_{N,\Gamma_{E_8}}$ | $16N$ | $334N-248$ | $12N-1$ |

Table 2: Anomaly coefficients $\kappa_{1,2,3}$ for an assortment of 6d SCFTs I. We exhibit both the values computed at the SCFT point using the formula (1.16), and the naive values by simply adding free fields on the tensor branch.

| | Coefficient | SCFT | Naive tensor branch | Difference |
|---|---|---|---|---|
| Free theory | $\kappa_1$ | $-n_H+n_V+n_T$ | $-n_H+n_V+n_T$ | 0 |
| | $\kappa_2-\frac{3}{2}\kappa_3$ | $\frac{1}{2}(n_H-n_V+5n_T)$ | $\frac{1}{2}(n_H-n_V+5n_T)$ | 0 |
| | $\kappa_3$ | $n_V-n_T$ | $n_V-n_T$ | 0 |
| | $\kappa_\text{f}^{USp(2n_H)}$ | $-1$ | $-1$ | 0 |
| E-string | $\kappa_1$ | $-30N+1$ | 1 | $-30N$ |
| | $\kappa_2-\frac{3}{2}\kappa_3$ | $15N-\frac{1}{2}$ | $3N-\frac{1}{2}$ | $12N$ |
| | $\kappa_3$ | $N(6N+5)$ | $-N$ | $6N(N+1)$ |
| | $\kappa_\text{f}^{E_8}$ | $-12N$ | 0 | $-12N$ |
| type-$\mathfrak{g}$ $(2,0)$ | $\kappa_1$ | 0 | 0 | 0 |
| | $\kappa_2-\frac{3}{2}\kappa_3$ | $3r_\mathfrak{g}$ | $3r_\mathfrak{g}$ | 0 |
| | $\kappa_3$ | $-r_\mathfrak{g}$ | $-r_\mathfrak{g}$ | 0 |
| | $\kappa_\text{f}^{SU(2)}$ | $-r_\mathfrak{g}$ | $-r_\mathfrak{g}$ | 0 |

In general a subgroup $H$ of the UV flavor symmetry group $G$ acts nontrivially on the hypermultiplets, and it is embedded into the $USp(2n_H)$ symmetry. The $\kappa_\text{f}^G$ can be computed by the formulae

$$\begin{aligned}
\kappa_\text{f}^H &= \mathcal{I}_{H\hookrightarrow G}\kappa_\text{f}^G\,,\\
\kappa_\text{f}^H &= \mathcal{I}_{H\hookrightarrow USp(2n_H)}\kappa_\text{f}^{USp(2n_H)} = -\mathcal{I}_{G\hookrightarrow USp(2n_H)}\,,
\end{aligned} \qquad (3.24)$$

where $\mathcal{I}_{H\hookrightarrow G}$ and $\mathcal{I}_{H\hookrightarrow USp(2n_H)}$ are the embedding indices of the embedding of the UV flavor subgroup $H$ into the UV flavor group or the $USp(2n_H)$ symmetry or $n_H$ free hypermultiplets.

For the E-string, $\mathcal{T}_{N,E_6}$, $\mathcal{T}_{N,E_7}$, and $\mathcal{T}_{N,E_8}$ theories, the UV flavor group $G$ does not act on the hypermultiplets, i.e. the subgroup $H$ is trivial. Hence, the Chern-Simons level $\kappa_\text{f}$ for those theories on the tensor branches is zero.

For the type-$\mathfrak{g}$, $\mathcal{N}=(2,0)$ theories viewed as $\mathcal{N}=(1,0)$ theories, the UV $SU(2)$ symmetry is embedded as the diagonal subgroup of $USp(2)^{r_\mathfrak{g}} \subset USp(2r_\mathfrak{g})$. The embedding index is

$$\mathcal{I}_{SU(2)\hookrightarrow USp(2)^{r_\mathfrak{g}}} = r_\mathfrak{g}\,, \qquad \text{and} \qquad \mathcal{I}_{USp(2)^{r_\mathfrak{g}}\hookrightarrow USp(2r_\mathfrak{g})} = 1\,. \qquad (3.25)$$

Table 3: Anomaly coefficients $\kappa_{1,2,3}$ for an assortment of 6d SCFTs II. We exhibit both the values computed at the SCFT point using the formula (1.16), and the naive values by simply adding free fields on the tensor branch.

| | Coefficient | SCFT | Naive tensor branch | Difference |
|---|---|---|---|---|
| $\mathcal{T}_{N,\Gamma_{A_{k-1}}}$ | $\kappa_1$ | $-k^2$ | $-k^2$ | $0$ |
| | $\kappa_2 - \frac{3}{2}\kappa_3$ | $\frac{1}{2}\left(k^2+6N-6\right)$ | $\frac{1}{2}\left(k^2+6N-6\right)$ | $0$ |
| | $\kappa_3$ | $\left(k^2-2\right)(N-1)$ | $\left(k^2-2\right)(N-1)$ | $0$ |
| | $\kappa_{\mathrm{f}}^{SU(k)_L}$ or $\kappa_{\mathrm{f}}^{SU(k)_R}$ | $-2k$ | $-2k$ | $0$ |
| $\mathcal{T}_{N,\Gamma_{D_k}}$ | $\kappa_1$ | $-2k^2+k-1$ | $-2k^2+k+30N-1$ | $-30N$ |
| | $\kappa_2 - \frac{3}{2}\kappa_3$ | $k^2-\frac{k}{2}+3N-\frac{5}{2}$ | $k^2-\frac{k}{2}-9N-\frac{5}{2}$ | $12N$ |
| | $\kappa_3$ | $\begin{array}{c}k^2(4N-2)-4kN\\+k-10N+1\end{array}$ | $\begin{array}{c}k^2(4N-2)-16kN\\+k+26N+1\end{array}$ | $12(k-3)N$ |
| | $\kappa_{\mathrm{f}}^{SO(2k)_L}$ or $\kappa_{\mathrm{f}}^{SO(2k)_R}$ | $4-4k$ | $16-4k$ | $-12$ |
| $\mathcal{T}_{N,\Gamma_{E_6}}$ | $\kappa_1$ | $-79$ | $90N-79$ | $-90N$ |
| | $\kappa_2 - \frac{3}{2}\kappa_3$ | $3N+\frac{73}{2}$ | $-33N+\frac{73}{2}$ | $36N$ |
| | $\kappa_3$ | $166N-77$ | $82N-77$ | $84N$ |
| | $\kappa_{\mathrm{f}}^{(E_6)_L}$ or $\kappa_{\mathrm{f}}^{(E_6)_R}$ | $-24$ | $0$ | $-24$ |
| $\mathcal{T}_{N,\Gamma_{E_7}}$ | $\kappa_1$ | $-134$ | $150N-134$ | $-150N$ |
| | $\kappa_2 - \frac{3}{2}\kappa_3$ | $3N+64$ | $-57N+64$ | $60N$ |
| | $\kappa_3$ | $382N-132$ | $154N-132$ | $228N$ |
| | $\kappa_{\mathrm{f}}^{(E_7)_L}$ or $\kappa_{\mathrm{f}}^{(E_7)_R}$ | $-36$ | $0$ | $-36$ |
| $\mathcal{T}_{N,\Gamma_{E_8}}$ | $\kappa_1$ | $-249$ | $330N-249$ | $-330N$ |
| | $\kappa_2 - \frac{3}{2}\kappa_3$ | $3N+\frac{243}{2}$ | $-129N+\frac{243}{2}$ | $132N$ |
| | $\kappa_3$ | $1078N-247$ | $322N-247$ | $756N$ |
| | $\kappa_{\mathrm{f}}^{(E_8)_L}$ or $\kappa_{\mathrm{f}}^{(E_8)_R}$ | $-60$ | $0$ | $-60$ |

We find the Chern-Simons level

$$\kappa_{\mathrm{f}}^{SU(2)} = -r_{\mathfrak{g}}, \tag{3.26}$$

for the type-$\mathfrak{g}$ $\mathcal{N}=(2,0)$ theories on the tensor branch.

For $\mathcal{T}_{N,\Gamma_{A_{k-1}}}$, there are $k^2 N$ hypermultiplets on the tensor branch. Only $k^2$ of them are charged under the UV flavor symmetry $SU(k)_L$. In the IR limit, the $k^2$ free hypermultiplets transform under $USp(2k^2)$. The UV $SU(k)_L$ symmetry embeds into the IR $USp(2k^2)$ symmetry as $SU(k) \hookrightarrow SU(k)\times SU(k) \hookrightarrow SU(k^2) \hookrightarrow USp(2k^2)$. We find the following embedding indices

$$\mathcal{I}_{SU(k)\hookrightarrow SU(k)\times SU(k)} = k, \quad \mathcal{I}_{SU(k)\times SU(k)\hookrightarrow SU(k^2)} = 1, \quad \mathcal{I}_{SU(k^2)\to USp(2k^2)} = 2. \tag{3.27}$$

Therefore, the Chern-Simons levels $\kappa_{\mathrm{f}}^{SU(k)_L}$ and $\kappa_{\mathrm{f}}^{SU(k)_R}$ on the tensor branch are

$$\kappa_{\mathrm{f}}^{SU(k)_L} = \kappa_{\mathrm{f}}^{SU(k)_R} = -2k. \tag{3.28}$$

For $\mathcal{T}_{N,\Gamma_{D_k}}$, there are $2k(2k-8)N$ hypermultiplets on the tensor branch, but only $k(2k-8)$ of them are charged under the UV flavor symmetry $SO(2k)_L$. In the IR limit, the $k(2k-8)$ free hypermultiplets transform under $USp(2k(2k-8))$. The UV $SO(2k)_L$ symmetry embeds into

the IR $USp(2k(2k-8))$ symmetry as $SO(2k) \hookrightarrow SO(2k) \times USp(2k-8) \hookrightarrow USp(2k(2k-8))$. We find the embedding indices

$$\mathcal{I}_{SO(2)\hookrightarrow SO(2k)\times USp(2k-8)} = 2k-8, \quad \mathcal{I}_{SO(2k)\times USp(2k-8)\hookrightarrow USp(2k(2k-8))} = 2. \tag{3.29}$$

Therefore, the Chern-Simons levels $\kappa_{\mathrm{f}}^{SO(2k)_L}$ and $\kappa_{\mathrm{f}}^{SO(2k)_R}$ on the tensor branch are

$$\kappa_{\mathrm{f}}^{SO(2k)_L} = \kappa_{\mathrm{f}}^{SO(2k)_R} = 16-4k. \tag{3.30}$$

# 4 Chern-Simons levels and global anomalies

There is a direct relation between the Chern-Simons levels $\kappa_i$ in the 5d effective action and the phases from global anomalies in the 6d theory [48, 49, 97]. This connection immediately implies that the Chern-Simons levels $\kappa_i$ can only jump by an appropriate quantized amount along renormalization group flows into the Higgs or tensor branch moduli spaces. This supports the argument of Section 1.4 that $\kappa_i$ are in fact constant on the entire vacuum moduli space.

We put the theory on a six manifold $\mathcal{M}_6$ (1.22) that is an $S^1$ fibration over a five manifold $\mathcal{M}_5$, and study the large diffeomorphisms of the coordinate $\tau$ of the $S^1$ fiber. The coordinate $\tau$ has periodicity $\beta$, *i.e.* $\tau \sim \tau + \beta$. The boundary condition for the fermionic degrees of freedom along the $S^1$ fiber is chosen to be periodic to preserve supersymmetry. Due to this boundary condition, the fermionic degrees of freedom on $\mathcal{M}_5$ have integer charges under the background graviphoton,

$$\frac{1}{2\pi} \int_{\Sigma_2} dA \in \mathbb{Z}, \tag{4.1}$$

where $\Sigma_2$ is a two-cycle in $\mathcal{M}_5$. Therefore, the five manifold $\mathcal{M}_5$ must be a spin manifold.[39]

In the low energy limit, the tensor or Higgs branch effective theories always contain free fermions. For the partition function on $\mathcal{M}_6$ to be nonzero, we need to choose a background such that there is no fermionic zero mode. Let us now assume that $\mathcal{M}_5 = S^1_{x_5} \times \mathcal{M}_4$, with the radius of $S^1_{x_5}$ being $R_5$, *i.e.*

$$x_5 \sim x_5 + 2\pi R_5. \tag{4.3}$$

The boundary condition of the fermionic degrees of freedom on $S^1_{x_5}$ is chosen to be periodic. The fermion zero modes of the IR free fermions can be lifted by turning on a nontrivial flat connection of either the R-symmetry or flavor symmetry background gauge fields along the $x_5$ direction. This is equivalent to trivial background gauge fields but with nontrivial boundary conditions for the charged degrees of freedom.

Consider a background diffeomorphism as follows

$$\tau \rightarrow \tau + \frac{n\beta}{2\pi R_5} x^5, \tag{4.4}$$

---

[39] If the fermionic degrees of freedom have antiperiodic boundary conditions along the $S^1$, then after the dimensional reduction the fermionic degrees of freedom on $\mathcal{M}_5$ would have half-integer charges under the background $U(1)_{\mathrm{KK}}$ graviphoton gauge field $A$ given in the metric (1.22). The five manifold $\mathcal{M}_5$ could be a more general spin$_c$ manifold. More precisely, the fermionic degrees of freedom on $\mathcal{M}_5$ have integer charges under a spin$_c$ connection $A_c = \frac{1}{2}A$, which satisfies

$$\frac{1}{2\pi} \int_{\Sigma_2} dA_c = \frac{1}{2} \int_{\Sigma_2} w_2 \mod \mathbb{Z}, \tag{4.2}$$

where $w_2$ is the second Stiefel-Whitney class of $\mathcal{M}_5$.

where $n \in \mathbb{Z}$ to preserve the boundary conditions for the fermionic degrees of freedom along the $S^1_{x_5}$. The background diffeomorphism corresponds to the background large gauge transformation of the graviphoton $A$,

$$A \rightarrow A + \frac{n}{R_5} dx^5 \,. \tag{4.5}$$

In general, for theories with (mixed) gravitational anomalies, the partition function is not invariant under such a background diffeomorphism, and transforms by a phase,

$$Z[A + \delta A] = e^{-i\pi\eta} Z[A] \,. \tag{4.6}$$

The 5d effective action completely captures this anomalous diffeomorphism. Under the large gauge transformation (4.5), the effective action transforms as

$$\delta W = n \int_{\mathcal{M}_4} \left( \frac{\kappa_1}{480\pi} dA \wedge dA - \frac{\pi}{72}(\kappa_2 - \frac{3}{2}\kappa_3) p_1 - \frac{\pi}{12}\kappa_3 c_2(SU(2)_R) - \frac{\pi}{12}\kappa_f^{G_f} c_2(G_f) \right) \,. \tag{4.7}$$

The above integral satisfies quantization conditions given by various index theorems on the spin manifold $\mathcal{M}_4$

$$
\begin{aligned}
m_1 &\equiv \frac{1}{2(2\pi)^2} \int_{\mathcal{M}_4} dA \wedge dA \in \mathbb{Z} \,, \\
m_2 &\equiv \frac{1}{24} \int_{\mathcal{M}_4} p_1 \in 2\mathbb{Z} \,, \\
m_3 &\equiv \int_{\mathcal{M}_4} c_2(SU(2)_R) \in \mathbb{Z} \,, \\
m_f &\equiv \int_{\mathcal{M}_4} c_2(G_f) \in \mathbb{Z} \,.
\end{aligned}
\tag{4.8}
$$

We find that the anomalous phase is given by

$$\eta = \frac{nm_1}{60}\kappa_1 + \frac{2nm_2}{3}(\kappa_2 - \frac{3}{2}\kappa_3) - \frac{n}{12}(m_3\kappa_3 + m_f\kappa_f^{G_f}) \mod 2 \,. \tag{4.9}$$

From the fifth columns of Table 2 and 3, we find explicitly the mismatches between the global anomaly $\eta$ of the UV SCFT (computed using the formula (1.16)) and the IR effective theory on the tensor branch. However, anomaly matching on the tensor branch is more difficult due to the (possible) subtle contributions of Green-Schwarz type terms which are generated from integrating out massive BPS strings. Recent progress on understanding the contribution of the Green-Schwarz type terms to global anomalies has been made in [80–86]. For theories that also have a Higgs branch (where the formula (1.16) was proven), anomaly matching between the two branches then predicts what the Green-Schwarz type contributions must be.

## 5 Conclusion

We have proven the universality of the Cardy limit of the superconformal index for 6d SCFTs with a pure Higgs branch, embodied in a precise formula (1.14) conjectured by Di Pietro and Komargodski in [1] for the singular terms, with coefficients related to the perturbative anomalies via (1.16).

**Summary of proof**

i) Compactifying the 6d SCFT on $S^1_\beta$, in the Cardy limit ($\beta \to 0$), one obtains the 5d effective action (1.20) that contains four supersymmetric Chern-Simons terms $I_j$, for $j = 1, \ldots 4$, with explicit expressions given in (1.21).

ii) We evaluate $\{I_j\}^4_{j=1}$ on the supersymmetric squashed $S^5$ background, and determine the explicit dependence on the squashing parameters $\omega_i$. The dependence is in agreement with the proposal (1.14). The pieces that supersymmetrically complete the Chern-Simons terms contribute nontrivially; in fact, their contributions are crucial for the final answer to be a geometric invariant, *i.e.* dependent only on the transverse holomorphic structure.

iii) To prove the relation (1.16) between the Chern-Simons levels $\kappa_j$ and the coefficients $\alpha, \beta, \gamma$ and $\delta$ in the 8-form anomaly polynomial (1.15), the following are the steps.

    a) First, we use background infinitesimal gauge invariance to argue that the Chern-Simons levels $\kappa_j$ are invariant under RG flows of the 6d SCFT on its moduli space.

    b) The IR effective theory on the Higgs branch is a theory of hypermultiplets (free at large moduli), whose field content can be determined explicitly from the generic (global) symmetry breaking pattern along the RG flow.

    c) We explicitly KK-reduce the free hypermultiplets along the $S^1_\beta$, and determine the Chern-Simons levels $\kappa_j$ by integrating out massive KK modes at one-loop. The result can be expressed in terms of the perturbative anomaly coefficients $\alpha, \beta, \gamma$ and $\delta$ as in (3.22).

    d) Since $\kappa_i$, $\alpha, \beta, \gamma$ and $\delta$ are all invariant under the pure Higgs branch flow, the relation (3.22) holds at the UV superconformal fixed point as well.

**A puzzle on the tensor branch**

iv) By looking at various examples, we remarked that the invariance of the Chern-Simons levels along the tensor branch flow requires extra contributions to the $\kappa_i$'s, in addition to the one-loop contributions from the free fields. We leave a more in-depth study of these extra contributions for future work [98].[40]

**Global anomaly matching**

v) We relate the Chern-Simons couplings $\kappa_j$ to global gravitational anomalies. By explicitly studying the variation of the effective action under large diffeomorphisms, we derive a relation between the anomalous factor $\eta$ in (4.6) and the Chern-Simons levels $\kappa_j$.

**Future prospects**

An obvious avenue for exploration is to study more general 5d manifolds with non-trivial topology. This is especially interesting in view of the relation between the 5d Chern-Simons coefficients and global anomalies, the latter being sensitive to the topology of the space.

---

[40]We note that the full 6d anomaly polynomial for a large class of theories are not fully determined as of the writing of this paper.

The 2d Cardy formula is intimately related to modular properties of the torus partition function. In fact, modularity properties for $\mathcal{N} = (2,0)$ [74,76,79] have also been understood. It is an intriguing question to ask whether the Cardy formula for $\mathcal{N} = (1,0)$ theories also has a modular origin.

An alternative abstract way of determining the Chern-Simons levels for the 5d effective action in the Cardy limit was given in [99,100]. Their argument involves putting the CFT on conical geometries, and assuming that the partition function is well-defined. In particular, the answer should not depend on the different resolutions of the singularity. It would be interesting to extend their analysis to the supersymmetric setting, in which case the supersymmetric partition function is known to be insensitive to the resolutions of conical singularities [101–103], which may put their arguments on a more rigorous footing.

Lastly, we comment on the relation between the Cardy formula proven in this paper, and recent work relating the "Cardy limit" of superconformal indices in 4d and 6d to black holes entropies [5–8]. There is an ambiguity in nomenclature here. Whereas the Cardy limit considered in this paper fixes the squashing parameters, the limit considered in [5–7] has the (complexified) squashing parameters scale with $\beta$. The fugacities $\omega_i$ in [5–8] are related to our notation by

$$\beta \omega_1^{(\text{here})} = \omega_1^{(\text{there})}, \quad \beta \omega_2^{(\text{here})} = \omega_2^{(\text{there})}, \quad \beta \omega_3^{(\text{here})} = 2\pi i + \omega_3^{(\text{there})}, \tag{5.1}$$

and the "Cardy limit" defined in [5–7] is the small $\omega_i^{(\text{there})}$ limit. The terms in the 5d effective action that dominate in this new limit are *different* from our limit – additional terms that are non-invariant under background perturbative gauge transformations must be included.[41] Since this new limit appears important for black hole entropy-matching, it would be interesting to prove the universal relation between this new limit and the anomalies, by similar arguments as in the present work applied to a *different* 5d squashed sphere background. As a matter of fact, we show in Appendix C, that the squashed $S^5$ background for the "modified 6d index" can be obtained by a simple shift of $\omega_3$. Nonetheless, it remains to understand the supersymmetric completion of the additional pieces appearing in the corresponding effective action.[42]

## Acknowledgments

We are grateful to Lorenzo Di Pietro for extremely helpful correspondences, and are deeply indebted to Yuji Tachikawa for patiently explaining the various subtleties of global anomaly matching in 6d. We also thank Zohar Komargodski, Kantaro Ohmori, and Amos Yarom for insightful discussions. Finally, we thank Lorenzo Di Pietro, Zohar Komargodski, and Yuji Tachikawa for discerning comments on the draft. CC is supported in part by the U.S. Department of Energy grant DE-SC0009999. The work of MF is supported by the SNS fellowship P400P2-180740, the Princeton physics department, the JSPS Grant-In-Aid for Scientific Research Wakate(A) 17H04837, the WPI Initiative, MEXT, Japan at IPMU, the University of Tokyo, the David and Ellen Lee Postdoctoral Scholarship. YL is supported by the Sherman Fairchild Foundation, and both MF and YL by the U.S. Department of Energy, Office of Science, Office of High Energy Physics, under Award Number DE-SC0011632. YW is supported

---

[41]The present authors find no clear relation between the physical 5d effective action and the "5d effective action" in [8]. In particular, the effective action in [8] is not supersymmetric.

[42]The effective action in the limit of the modified index (see [8]) contains singular gauge non-invariant pieces as well as gauge-invariant ones. The supersymmetric completions of the gauge-invariant terms have been considered in the present paper, and to study the limit of the modified index we simply need to evaluate them on the modified 5d background (see Appendix C). However, the supersymmetric completions of the gauge non-invariant terms are (to the knowledge of the authors) not known to date.

in part by the US NSF under Grant No. PHY-1620059 and by the Simons Foundation Grant No. 488653. YL and YW thank the Bootstrap 2019 conference, and CC, MF and YW thank the Aspen Center for Physics, which is supported by National Science Foundation grant PHY-1607611, for hospitality during the finishing stages of this paper.

# A  Conventions for characteristic classes

We denote by $p_1$ and $p_2$ the first and second Pontryagin classes, and $c_1(U(1))$ and $c_2(G)$ the first and second Chern class of the groups $U(1)$ and $G$, respectively. They can be written in terms of the corresponding curvatures as follows:

$$
\begin{aligned}
p_1 &= -\frac{1}{2(2\pi)^2}\text{tr}R^2\,,\\
p_2 &= \frac{1}{(2\pi)^4}\left[-\frac{1}{4}\text{tr}R^4 + \frac{1}{8}(\text{tr}R^2)^2\right],\\
c_1(U(1)) &= \frac{1}{2\pi}F_{U(1)}\,,\\
c_2(G) &= \frac{1}{2(2\pi)^2}\text{Tr}F_G^2\,.
\end{aligned}
\tag{A.1}
$$

Here, the trace tr is over the vector representation, and the trace Tr is defined as

$$
\text{Tr}(\cdot) \equiv \frac{1}{2h^\vee}\text{tr}_{\text{adj}}(\cdot)\,,
\tag{A.2}
$$

where $h^\vee$ is the dual Coxeter number.

# B  Central extensions of supersymmetry algebras and (extended) BPS objects for 4d $\mathcal{N} = 2$ and 6d $\mathcal{N} = (1,0)$ theories

In this Appendix, we provide a brief classification of BPS objects appearing in Higgs/Coulomb/mixed branch flows for 4d $\mathcal{N} = 2$ theories and tensor/Higgs/mixed branch flows for 6d $\mathcal{N} = (1,0)$ theories.

## B.1  4d $\mathcal{N} = 2$

The BPS states of 4d $\mathcal{N} = 2$ theories can be classified by looking at the central extensions of the supersymmetry algebra,

$$
\begin{aligned}
\{Q_\alpha^i, Q_\beta^j\} &= \epsilon_{\alpha\beta}\epsilon^{ij}Z + \sigma_{\alpha\beta}^{\mu\nu}Z_{\mu\nu}^{(ij)}\,,\\
\{\bar{Q}_{\dot\alpha}^i, \bar{Q}_{\dot\beta}^j\} &= \epsilon_{\dot\alpha\dot\beta}\epsilon^{ij}\bar{Z} + \bar{\sigma}_{\dot\alpha\dot\beta}^{\mu\nu}\bar{Z}_{\mu\nu}^{(ij)}\,,\\
\{Q_\alpha^i, \bar{Q}_{\dot\beta}^j\} &= \sigma_{\alpha\dot\beta}^\mu(\epsilon^{ij}(P_\mu + Z_\mu) + Z_\mu^{(ij)})\,.
\end{aligned}
\tag{B.1}
$$

Here, $Z$ is the usual central charge for BPS particles, whereas $Z_\mu, Z_\mu^{(ij)}$ and $Z_{\mu\nu}^{(ij)}$ are the brane charges for BPS strings and domain-walls, respectively.[43] In particular, the brane charges are

---

[43]Equivalently, one can study non-conformal modifications of the conformal supercurrent multiplets which contain additional brane currents, or non-conformal extensions of the conformal supergravity by introducing compensator multiplets (for reviews see [104–106]).

related to the (higher-form) brane currents for a $p$-form symmetry by

$$Z^{(p)}_{\mu_1\mu_2...\mu_p} \equiv T^{(p)} \int \mathrm{d}^{d-1}x\, J^{(p)}_{0\mu_1\mu_2...\mu_p}, \tag{B.2}$$

where the integral is over the spatial directions, and $T^{(p)}$ is the brane tension with mass dimension $p+1$ (since the current $J^{(p)}$ has dimension $d-p-1$). In terms of coupling to background non-conformal supergravity, the brane tension $T^{(p)}$ is given by the scalar vev of certain compensator multiplets. As argued in [105, 107], $Z_\mu$ should vanish for 4d $\mathcal{N}=2$ theories that come from deforming SCFTs, because they are expected to have Ferrara-Zumino (FZ) multiplets (with respect to any $\mathcal{N}=1$ subalgebra) [108].

As for the other brane charges that appear in the central extension of the 4d $\mathcal{N}=2$ supersymmetry algebra, the central charge $Z$ for BPS particles is given by the complex scalars in the vector multiplets, whereas the tensions for the BPS strings and domain walls also depend on the real $SU(2)_R$ triplet scalars in the linear multiplets. For effective theories on the moduli space of an SCFT, these tensions are determined by the vevs of vector multiplet and hypermultiplet scalars. In particular, the tension for a (unit-charge) BPS string is given by $q\tilde{q}$, where $q,\tilde{q}$ denote the chiral scalars in a hypermultiplet (this follows from the composite expression of a linear multiplet in terms of hypermultiplets). As for the BPS domain wall, the 3-form current is proportional to the Hodge dual of $\mathrm{d}\phi$ where $\phi$ is a complex scalar in a vector multiplet, and the tension for a BPS domain is given by $q\tilde{q}(\phi(+\infty)-\phi(-\infty))$, where the $q,\tilde{q}$ are hypermultiplet scalars charged under the vector multiplet.[44]

Consequently, the possible massive stable BPS states are:

(i) BPS particles on the Coulomb branch,

(ii) BPS strings on the Higgs branch,

(iii) BPS particles, strings and domain-walls on the mixed branches.

## B.2 6d $\mathcal{N}=(1,0)$

The BPS states of 6d $\mathcal{N}=(1,0)$ theories can be classified by looking at the central extensions of the supersymmetry algebra,

$$\{Q^i_\alpha, Q^j_\beta\} = \gamma^\mu_{\alpha\beta}\epsilon^{ij}(P_\mu + Z_\mu) + \gamma^{\mu\nu\rho}_{\alpha\beta} Z^{(ij)}_{\mu\nu\rho}. \tag{B.3}$$

Here, $Z_\mu$ is the string charge and $Z_{\mu\nu\rho}$ the 4-brane charge, and due to supersymmetry, the corresponding brane tensions are respectively given by the real scalar in the tensor multiplet and the real $SU(2)_R$-triplet scalars in the linear multiplet (which contains a 4-form gauge field that couples to the brane). By a similar reasoning as for 4d $\mathcal{N}=2$, the possible massive BPS states are

(i) BPS strings on the tensor branch,

(ii) BPS codimension-two vortex branes on the Higgs branch, and

(iii) BPS strings and vortex branes on the mixed branches.

Upon compactification on $S^1$ (and before compactifying the 5d theory on $S^5$), the (wrapped) vortex branes have infinite energy and thus do not contribute to the Chern-Simons levels in the 5d effective action. Thus, in conclusion, there are no massive BPS states contributing to the 5d effective action for Higgs branch flows.

---

[44]Note that in $\mathcal{N}=1$ notation, this is just the usual statement that domain wall tensions are determined by the difference between the values of the superpotential at adjacent vacua.

# C   Supersymmetric $S^1 \times S^5$ background for the index and modified index

This appendix discusses the supersymmetric background for 6d theories on $S^1 \times S^5$ twisted by chemical potentials. The notation used in this section is independent of the main text.

We start with the 6d metric in equation (2.1), where we set $r_5 = 1$, and pick the following frame

$$
\begin{aligned}
e^1 &= \mathrm{d}\tau\,, & e^2 &= \mathrm{d}\theta\,, \\
e^3 &= \sin\theta\,\mathrm{d}\psi\,, & e^{j+3} &= y_j(\mathrm{d}\phi_j + \mathrm{i}a_j\mathrm{d}\tau)\,, \ j = 1,2,3\,,
\end{aligned}
\tag{C.1}
$$

where we have introduced the spherical coordinates

$$
y_1 = \sin\theta\cos\psi\,, \qquad y_2 = \sin\theta\sin\psi\,, \qquad y_3 = \cos\theta\,,
\tag{C.2}
$$

with $\psi \in [0, 2\pi)$ and $\theta \in [0, \pi)$.

In order to preserve some supersymmetry, we couple the theory to 6d background (off-shell) conformal supergravity. More precisely, we couple to the "6d Weyl 1 multiplet", also known as the "6d standard Weyl multiplet" [109–111],[45] which contains the bosonic fields given by the metric, $g_{\mu\nu}$, the gauge field for Weyl transformations $b_\mu$, an $\mathfrak{su}(2)_R$ symmetry gauge field $V_\mu{}^i{}_j$, an antisymmetric tensor $T^-_{abc}$ and finally a real scalar $D$ as well as fermionic fields, $\psi^i_\mu$, $\chi^i$ $\mathfrak{su}(2)_R$ Majorana-Weyl spinors of positive, negative chirality, respectively.

We are interested in the rigid limit [36], and thus set the fermionic fields to zero. Furthermore, to preserve some SUSY, we have to find non-trivial Killing spinors $\varepsilon^i$ and their conformal cousins, $\eta^i$, such that the variations of the fermionic fields vanish. The first constraint, $\delta\chi^i = 0$, is automatically solved by setting

$$
T^-_{abc} = D = 0\,.
\tag{C.3}
$$

With this, the remaining supersymmetry condition is then given by

$$
\delta\psi^i_\mu = \partial_\mu\varepsilon^i + \frac{1}{4}\omega_\mu{}^{ab}\Gamma_{ab}\varepsilon^i + V_\mu{}^i{}_j\varepsilon^j + \Gamma_\mu\eta^i = 0\,.
\tag{C.4}
$$

In the following, we pick the 6d Gamma matrices as follows

$$
\Gamma_i = \mathrm{i}\sigma_2 \otimes \gamma_i \quad (i = 1,\ldots,5)\,, \qquad \Gamma_6 = \sigma_1 \otimes \mathbb{1}_{4\times 4}\,,
\tag{C.5}
$$

where we recall that by $\gamma_i$ are the 5d Gamma matrices, defined in equation (2.25). We may solve (C.4) by turning on a background $\mathfrak{u}(1)_R \subset \mathfrak{su}(2)_R$ gauge-field,

$$
V_\mu{}^i{}_j\mathrm{d}x^\mu = \left(1 + \frac{V_0}{2}\right)\mathrm{d}\tau\,(\sigma_3)^i{}_j\,.
\tag{C.6}
$$

We explicitly find the following solutions for the 6d Killing spinors $\varepsilon^i$ and their conformal

---

[45]This is to make a distinction from another 6d $\mathcal{N} = (1,0)$ Weyl multiplet – "6d Weyl 2 multiplet" or "dilaton Weyl multiplet" – which has the same gauge but different matter fields [109, 110].

cousins $\eta^i$:

$$
\varepsilon^j(\tau) = \sqrt{2}e^{-\frac{\tau}{2}(1+a_{\text{tot}}+(-1)^{j+1}(2+V_0))+\frac{i}{2}\phi_{\text{tot}}}
\begin{pmatrix}
\mathbf{0}_{4\times1} \\
i\sin\frac{\theta}{2}e^{i\frac{\psi}{2}} \\
-\sin\frac{\theta}{2}e^{-i\frac{\psi}{2}} \\
i\cos\frac{\theta}{2}e^{i\frac{\psi}{2}} \\
-\cos\frac{\theta}{2}e^{-i\frac{\psi}{2}}
\end{pmatrix},
$$

$$
\eta^j(\tau) = \frac{1}{\sqrt{2}}e^{-\frac{\tau}{2}(1+a_{\text{tot}}+(-1)^{j+1}(2+V_0))+\frac{i}{2}\phi_{\text{tot}}}
\begin{pmatrix}
-\sin\frac{\theta}{2}e^{i\frac{\psi}{2}} \\
-i\sin\frac{\theta}{2}e^{-i\frac{\psi}{2}} \\
\cos\frac{\theta}{2}e^{i\frac{\psi}{2}} \\
i\cos\frac{\theta}{2}e^{-i\frac{\psi}{2}} \\
\mathbf{0}_{4\times1}
\end{pmatrix},
\tag{C.7}
$$

where $\phi_{\text{tot}} = \sum_{i=1}^3 \phi_i$ and similarly $a_{\text{tot}} = \sum_{i=1}^3 a_i$, and with $\mathbf{0}_{4\times1}$ a column vector with four zeroes as entries.

So far we have effectively been dealing with $\mathbb{R}\times S^5$. Now, we compactify the $\tau$-direction, and require the Killing spinors to satisfy the following consistency condition as we go around the $\tau$-circle

$$
\varepsilon^j(\tau+\beta) \equiv e^{\pi i n}\varepsilon^j(\tau), \qquad n\in\{0,1\},
\tag{C.8}
$$

where $n=0$ gives us back the usual index as defined in (1.13), while $n=1$ gives us the background for the modified index [5,112].[46] Therefore, we arrive at the conditions

$$
V_0 + \omega_1 + \omega_2 + \omega_3 = \frac{2\pi i n}{\beta},
\tag{C.9}
$$

where $\omega_i = 1 + a_i$. The analogous constraint in 4d was found in [112] by considering the 4d background $S^1\times S^3$ with modified periodicity for the fermions along the $S^1$.

To go from the usual index (*i.e.* $n=0$ in (C.9)) considered throughout the present paper to the modified index (*i.e.* $n=1$ in (C.9)), we can simply shift (*e.g.*) $\omega_3$, *i.e.*

$$
\omega_3 \longrightarrow \omega_3 + \frac{2\pi i}{\beta}.
\tag{C.10}
$$

Applying this shift to the evaluation of the supersymmetric completions of the higher-derivative terms (2.51), we find the following answers for the modified background:

$$
\begin{aligned}
I_{1,n=1}^{\text{mod}} &= -\frac{i}{\omega_1\omega_2}\frac{(2\pi)^5}{\beta^2} + \frac{\omega_3}{\omega_1\omega_2}\frac{(2\pi)^4}{\beta} + \frac{i\omega_3^2}{\omega_1\omega_2}(2\pi)^3 + \mathcal{O}(\beta^1), \\
I_{2,n=1}^{\text{mod}} &= \frac{2i}{\omega_1\omega_2}\frac{(2\pi)^5}{\beta^2} + \frac{2\omega_3}{\omega_1\omega_2}\frac{(2\pi)^4}{\beta} - \frac{2i(2\pi)^3\left(\omega_1^2+\omega_2^2\right)}{\omega_1\omega_2} + \mathcal{O}(\beta^1), \\
I_{3,n=1}^{\text{mod}} &= -\frac{i\pi^2}{\omega_1\omega_2}\frac{(2\pi)^3}{\beta^2} - \frac{\pi^2\left(2\omega_1+2\omega_2+\omega_3\right)}{\omega_1\omega_2}\frac{(2\pi)^2}{\beta} \\
&\quad + \frac{2i\pi^3\left(\omega_1^2+2\omega_2\omega_1+\omega_2^2\right)}{\omega_1\omega_2} + \mathcal{O}(\beta^1), \\
I_{4,n=1}^{\text{mod}} &= -\frac{i(2\pi)^3(m_{\text{f}}^I)^2}{\omega_1\omega_2} + \mathcal{O}(\beta^1).
\end{aligned}
\tag{C.11}
$$

---

[46]See [57], where a similar modification has originally been considered in the $\mathcal{N}=2$ superconformal Schur index.

We remark that this is the answer (together with the constraint (C.9)) in the strict $\beta \to 0$ limit. In fact, the constraint is crucial to render the result in (C.11) invariant under the exchange of $\omega_i$. Finally, in order to move to the "new Cardy limit" [5], we need to further rescale

$$\beta \omega_i \to \omega_i,\tag{C.12}$$

and then take the $\omega_i \to 0$ limit. However, as mentioned in the main text, the background gauge-invariant supersymmetric Chern-Simons terms treated in this paper are not the complete set of terms contributing to that limit.

# D  Geometric invariance and the equivalence between gauges

In the main text, we introduced two different solutions for background vector multiplets coupled to the standard Weyl multiplet: one was the "flavor solution" (2.26), and the other was the "KK solution" (2.27), where the gauge field is naturally identified with the 6d graviphoton. Accordingly, we introduced two distinct gauge-fixed version of Poincaré supergravity: the "standard gauge-fixed" (2.16) and the "KK gauge-fixed" (2.17) one. As one can explicitly confirm, the various types of solutions (in different combinations) lead to the *exact* same results for the evaluation of the supersymmetric Chern-Simons terms, giving more credence to the expectation that they are geometric invariants, *i.e.* only dependent on some simple geometric properties, believed to be the transversally holomorphic foliation inherent in the $\mathfrak{u}(1)_R$ truncated rigid supersymmetric backgrounds [14]. In this appendix, we provide a relation between the various solutions.

Let us first focus on the vector multiplet. A vector multiplet coupled to the standard Weyl multiplet is conformal. The (bosonic) fields in the standard Weyl and vector multiplet transform under Weyl rescaling as follows: (for completeness, we added the Weyl transformation of the linear multiplet)

$$
\begin{aligned}
\mathcal{SW}: \quad & g & \to \Lambda^2 g\,, \quad & V^i{}_j \to & V^i{}_j\,, \quad & D & \to \Lambda^{-2} D\,, \quad & v \to & \Lambda^2 v\,, \\
\mathcal{V}: \quad & M & \to \Lambda M\,, \quad & W_\mu \to & W_\mu\,, \quad & (Y)^i{}_j & \to \Lambda^2 (Y)^i{}_j\,, \\
\mathcal{L}: \quad & L^i{}_j & \to \Lambda^3 L^i{}_j\,, \quad & E_\mu \to & \Lambda^3 E_\mu\,, \quad & N & \to \Lambda^4 N\,.
\end{aligned}
\tag{D.1}
$$

From these expressions, it is clear that starting with the KK solution, we can fix the scalar $M$ to be a constant by $\Lambda = \mu_{\mathrm{f}} M^{-1}$, where $m$ is the constant flavor mass. However, this would change the metric, which is not what we want. Instead, we may connect the flavor solution to the KK solution by Weyl transforming the standard Weyl solution (2.21) according to the first line in (D.1) with the explicit Weyl factor

$$\Lambda = 2\tilde{\kappa} + \frac{1}{\tilde{\beta}\tilde{\kappa}}\,.\tag{D.2}$$

Then, one explicitly finds

$$\mathrm{d} W_{\mathrm{KK}} = \mathrm{d}\big[\mathrm{w}(W_{\mathrm{f}})\big] + \frac{4}{\tilde{\beta}}(e^1 \wedge e^2 + e^3 \wedge e^4)\,,\tag{D.3}$$

where by $\mathrm{w}(W_{\mathrm{f}})$ we mean the Weyl transform of the ingredients in the standard Weyl multiplet defining $\hat{W}$, *i.e.* in the constant-$M$ solution, $W_{\mathrm{f}} = -\mu_{\mathrm{f}}(1-(\tilde{\beta}\tilde{\kappa})^{-1})e^5$, which Weyl-transforms as follows[47]

$$\hat{W} \to \mathrm{w}(\hat{W}) = (1-(\Lambda\tilde{\beta}\tilde{\kappa})^{-1})(\Lambda e^5)\,.\tag{D.4}$$

---

[47]The reason for this curious Weyl transformation is that the ingredients in the (general) flavor solutions are actually bilinears in the Killing spinors, *i.e.* $W = (1-1/S)K_1$ with $K_1 \sim \epsilon^\dagger \gamma_{(1)} \epsilon$ and $S \sim \epsilon^\dagger \epsilon$ (see [14] for more details), which transform as in (3.8) of [14] under Weyl transformations.

Consequently, in the supersymmetry condition (2.12), part of the explicit $\Lambda$ in (D.2) will get absorbed by $Y$ and the $\tilde{\kappa}$ piece will be absorbed by $M$, making it non-constant. Thus, we move from the flavor solution with constant $M$ to a solution where $M$ is non-constant. Upon a (now) full Weyl transformation acting on the Standard Weyl as well as the vector multiplet back by

$$\Lambda = \frac{1}{2\tilde{\kappa} + \frac{1}{\tilde{\beta}\tilde{\kappa}}}, \tag{D.5}$$

we end up with the same metric but a different solution given by the KK solution.

By the same logic, one can relate the two gauge fixing conditions. These arguments then suggest that the resulting superconformal Chern-Simons terms, *i.e.* the FFF, $F_f F_f F$ and FWW terms, should be the same when evaluated on various solutions (which we explicitly checked), as they are by construction Weyl invariant. However, the reason for the FRR term, whose Weyl invariance is broken, to remain unchanged (explicitly checked) must be explained by the fact that it is a geometric invariant, *i.e.* solely depends on the transversely holomorphic structure of the solution.

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
