# Peer review of "Proving the 6d Cardy Formula and Matching Global Gravitational Anomalies"

_SciPost Physics, doi:SciPost Phys. 11, 036 (2021)_

## Round 1 · Referee Report · Anonymous (Referee 1) · 2021-5-31

Strengths

The paper is just right for the goals it aims to achieve. In particular,

1 - it is very clearly written;
2 - it contains a thorough analysis, at least for theories with a pure Higgs branch;
3 - an accurate but not exceedingly technical review of the needed background material is given; this is quite remarkable, given the vast and highly technical literature existing on the subject;
4 - the regime of applicability of the claims made is carefully explained.

Weaknesses

I do not see any particular weaknesses. If I really have to point my finger at something, I note that a few statements are justified via a reference to [88], which is work to appear by the same authors. My personal opinion is that it would be preferable to limit forward references to a minimum when they concern the proof of a statement (as they may leave a gap in the literature, at least for some time).

Report

The authors prove a Cardy formula for 6d SCFT's conjectured by Komargodski and Di Pietro in [1], where the small-$\beta$ limit of the partition function on $S^1_\beta \times S^5$ is given in terms of supersymmetric five-dimensional Chern-Simons terms and anomaly coefficients.
The proof is made of two main steps: $i)$ the identification of the relevant supersymmetric Chern-Simons terms in 5d off-shell supergravity, and their evaluation in the relevant background; $ii)$ the computation of the Chern-Simons levels.
The first part of the analysis involves mostly background off-shell supergravity and is quite thorough. The second part requires in principle a computation at the interacting IR fixed point and is challenging. The problem becomes easier for theories with a pure Higgs branch, where the coefficients of interest can be determined using a free theory. In this case the authors are able to complete their proof. The authors also pave the way for a more general analysis. In particular, they consider the case of the tensor branch, where they identify some interesting missing contributions from BPS strings whose detailed study is left for future work.
The paper is interesting and provides a systematic proof to the significant conjecture of [1].
I recommend publication in SciPost Physics with no hesitations.

Requested changes

The authors may specify if the twisting parameters $a_i$ appearing in the metric (2.1) and in (2.2) are real, imaginary, or generically complex.

  • validity: top
  • significance: high
  • originality: good
  • clarity: top
  • formatting: perfect
  • grammar: perfect

Author:  Chi-Ming Chang  on 2021-07-01  [id 1537]

(in reply to Report 1 on 2021-05-31)

We thank the referee for the comments. We will remove [88], and specify the reality property for the twisting parameters $a_i$ in the final published version.

---

## Round 1 · Referee Report · Lorenzo Di Pietro (Referee 2) · 2021-6-30

Strengths

1- Proof of the SUSY Cardy formula for non-Lagrangian theories using flow in the Higgs branch in 4d and 6d 2- Evaluation of the 5d supersymmetric Chern-Simons actions on the supersymmetric S^5 background 3- Observation that massive particles arising from BPS strings wrapping the circle contribute to the effective Chern-Simons term in Coulomb/Tensor branch flows

Weaknesses

1- The Cardy formula remains not proven in the general case with contributions from BPS strings

Report

The paper contains interesting results and is very well written so I recommend it for publications on SciPost.

---

## Round 2 · List of Changes

1. The reference [88] in the previous version removed.
2. A sentence below (2.2) added to specify the reality property of the twisting parameters.

---

## Editorial Decision

published